



# An index concentration method for suspended load monitoring in large rivers of the Amazonian foreland

William Santini[1,2], Benoît Camenen[3], Jérôme Le Coz[3], Philippe Vauchel[1,2], Jean-Loup Guyot[1,2], Waldo Lavado[4], Jorge Carranza[4], Marco A. Paredes[5], Jhonatan J. Pérez Arévalo[5], Nore Arévalo[6], Raul Espinoza Villar[6,7], Frédéric Julien[8], and Jean-Michel Martinez[1,2]

[1]IRD, Toulouse, 31400, France
[2]Laboratoire GET, CNRS, IRD, UPS, OMP, Toulouse, 31400, France
[3]IRSTEA, UR RIVERLY, Lyon-Villeurbanne, F-69625 Villeurbanne, France
[4]SENAMHI, Lima, Lima 11, Peru
[5]SENAMHI, Iquitos, Peru
[6]Facultad de Ingeniería Agrícola, UNALM, La Molina, Lima 12, Peru
[7]IGP, Ate, Lima 15012, Peru
[8]Laboratoire ECOLAB, CNRS, INPT, UPS, Toulouse, 31400, France

*Correspondence to*: William Santini (william.santini@ird.fr)

**Abstract.** Because increasing climatic variability and anthropic pressures have affected the sediment dynamics of large tropical rivers, long-term sediment concentration series have become crucial for understanding the related socio-economic and environmental impacts. For operational and cost rationalization purposes, index concentrations are often sampled in the flow and used as a surrogate of the cross-sectional average concentration. However, in large rivers where suspended sands are responsible for vertical concentration gradients, this index method can induce large uncertainties in the matter fluxes.

Assuming that physical laws describing the suspension of grains in turbulent flow are valid for large rivers, a simple formulation is derived to model the ratio ($\alpha$) between index and average concentrations. The model is validated using an exceptional dataset (1330 water samples, 249 concentration profiles, 88 particle size distributions (PSDs) and 494 discharge measurements) that was collected between 2010 and 2017 in the Amazonian foreland. The $\alpha$ prediction requires the estimation of the Rouse number ($P$), which summarizes the balance between the suspended particle settling and the turbulent lift, weighted by the ratio of sediment to eddy diffusivity ($\beta$). Two particle size groups, washload and sand, were considered to evaluate $P$. Discrepancies were observed between the evaluated and measured $P$, that were attributed to biases related to the settling and shear velocities estimations, but also to diffusivity ratios $\beta \neq 1$. An empirical expression taking into account these biases was then formulated to predict accurate estimates of $\beta$, then $P$ ($\Delta P = \pm 0.03$) and finally $\alpha$.

The proposed model is a powerful tool for optimizing the concentration sampling. It allows for detailed uncertainty analysis on the average concentration derived from an index method. Finally, this model can be coupled with remote sensing and hydrological modeling to serve as a step toward the development of an integrated approach for assessing sediment fluxes in poorly monitored basins.



## 1 Introduction

In recent decades, the Amazon basin has experienced an intensification in climatic variability (e.g. Gloor et al., 2013; Marengo et al., 2015) as well as increasing anthropic pressure. In the Peruvian foreland, the advance of the pioneer fronts causes serious changes in land use, which are enhanced by the proliferation of roads that provide access to the natural resources hosted by

this region. The number of hydropower projects is also rapidly increasing (e.g. Finer and Jenkins, 2012; Latrubesse et al., 2017; Forsberg et al., 2017).

These global and local changes might affect the sediment dynamics, generating large ecological impacts on the mega-diverse Amazonian biome and having socio-economic consequences on the riverine populations. In such a context, long-term and reliable sediment series are crucial for detecting, monitoring and understanding the related socio-economic and environmental

impacts (e.g. Walling, 1983, 2006; Horowitz, 2003; Syvitski et al., 2005; Horowitz et al. 2015). However, there is a lack of consistent data available for this region, and this lack of data has prompted an increased interest in developing better spatiotemporal monitoring of sediment transport.

In the large tropical rivers of Peru, the measurement of cross-sectional average concentrations $\langle C \rangle$ [mg L$^{-1}$] remains a costly and time-consuming task. First, gauging stations can only be reached after several days of travelling on hard dirt roads or by

the river. Second, there are no infrastructures on the rivers, and all operations are conducted using small boats under all flow conditions. Third, the gauging sections have depths that range from the metric to the decametric scale and widths that range from the hectometer to the kilometer scale. Such large sections experience pronounced sediment concentration gradients and grain size sorting, in both the vertical and the transverse directions (e.g. Curtis et al, 1979; Vanoni, 1979, 1980; Horowitz and Elrick, 1987; Filizola and Guyot, 2004; Filizola et al., 2010; Bouchez et al., 2011; Lupker et al., 2011; Armijos et al., 2013,

2016; Vauchel et al., 2017). The balance between the local hydrodynamic conditions and the sediment characteristics (e.g. grain size, density and shape) drive this spatial heterogeneity. Thus, the sand suspension is characterized by a high vertical gradient as well as a significant lateral variability and the concentration varies by several orders of magnitude; in contrast, the fine sediments (e.g. clays and silts) are transported homogeneously throughout the entire river section.

As a consequence, the entire cross-section must be explored to provide a representative estimate of the mean concentration of

coarse particles. Thus, it is necessary to identify a trade-off between the need to sample an adequate number of verticals and points throughout the cross-section and the need for time-integrated or repeated measurements to ensure the temporal representativeness of each water sample (Gitto et al., 2017). The second-order moments of the Navier-Stokes equations induce this temporal concentration variability, as do the larger turbulent structures (typically those induced by the bedforms) and the changes in flow conditions (e.g. backwaters, floods, and flow pulses). Sands and coarse silts are much more sensitive to

velocity fluctuations than are clay particles (i.e., settling laws are highly sensitive to the diameter of the particles) and are the most difficult to accurately measure.

Depth-integrated or point-integrated sampling procedures are traditionally used to determine the mean concentration of suspended sediment in rivers. However, deploying these methods from a boat is rarely feasible due to the velocity and depth





ranges that are encountered in large Amazonian rivers. For a point-integrated bottle sampling method, maintaining a position for a duration long enough to capture a representative water sample (Gitto et al., 2017) requires anchoring the boat and using a heavy ballast. This type of operation is very risky without good infrastructure and well-trained staff, especially when collecting measurements near the river's thalweg. Moreover, this method decreases the number of samples that can be collected

in one day.

For a depth-integrated sampling method within a deep river, the bottle may fill up before reaching the water surface if its transit speed is too slow. Moreover, if the ballast weight is not sufficient to hold the sampler nose in a horizontal position, the filling conditions are not isokinetic, and therefore, the sample will be non-representative.

Indirect surrogate technologies (e.g. laser diffraction technology or high-frequency acoustic instruments with multi-

transducers) may also be used. These instruments provide access to the temporal variability in concentration or grain size; however, they have limited ranges, post-processing complexity (Gray and Gartner, 2010; Armijos et al., 2016), and higher maintenance costs due the fragility of the instruments.

Thus, sampling methods with instantaneous capture or short-term integration ($< 30$ s) are preferred. These methods follow a relevant grid of sample points (Xiaoqing, 2003, Filizola and Guyot, 2004; Bouchez et al., 2011; Armijos et al., 2013; Vauchel

et al., 2017). The mean concentration $\langle C \rangle$ [mg L$^{-1}$] is determined by combining all samples into a single representative discharge-weighted concentration value, which is depth-integrated and cross-sectionally representative (Xiaoqing, 2003; Horowitz et al., 2015; Vauchel et al., 2017). In the present study, the spatial distribution of the concentration within the cross section is summarized into a single concentration profile that is assumed to be representative of the suspension regime along the river reach. The water depth $h$ [m] becomes the mean cross-section depth, which is close to the hydraulic radius for large

rivers. Therefore, $\langle C \rangle$ will be hereafter defined as the depth-integration of this concentration profile $C(z)$ ($z$ [m] being the height above the bed), from a reference height $z_0$ [m] just above the riverbed ($z_0 \ll h$) to the free surface, and weighted by the depth-averaged velocity $\langle u \rangle = \frac{1}{h} \int_{z_0}^{h} u(z)\, dz$ [m s$^{-1}$]:

$$\langle C \rangle = \frac{\int_{z_0}^{h} C(z) \times u(z)\, dz}{\int_{z_0}^{h} u(z)\, dz}, \tag{1}$$

However, to dampen the random uncertainties mainly related to the coarse sediments, this procedure requires taking a

statistically significant number of samples throughout the cross-section, which is also time- and labor-intensive.

All these limitations preclude the application of such complete sampling procedures at a relevant time-step necessary to build up a detailed concentration series.

By analogy with the index velocity method for discharge computation (Levesque and Oberg, 2012), derived surrogate procedures, called index sampling methods (Xiaoqing, 2003), are thus preferred. One or a few "index samples" are taken as

proxies of $\langle C \rangle$, usually at the water surface (e.g. Filizola and Guyot, 2004; Bouchez et al., 2011; Vauchel 2017). The index





concentration monitoring frequency is then scheduled to suit the river's hydrological behavior and minimize the random uncertainties on the measured index concentration $C(z_\chi)$ [mg L$^{-1}$] (Duvert et al., 2011; Horowitz et al., 2015).

The index concentration method first requires a robust site-specific calibration between the two concentrations of interest, $\langle C \rangle$ and $C(z_\chi)$, i.e., for all hydrological conditions, which cannot always be achieved under field conditions. Such relations are

usually expressed with the following linear form (e.g. Filizola, 2003; Guyot et al. 2005, 2007; Espinoza-Villar et al., 2012; Vauchel et al., 2017):

$$\langle C \rangle = \alpha \, C(z_\chi) + \xi \,, \tag{2}$$

where the regression slope $\alpha$ and the intercept $\xi$ are the fitted parameters of the empirical model. In this study, the intercept will be assumed to be zero ($\xi = 0$).

The dispersion and the extrapolation of the $\alpha = \langle C \rangle / C(z_\chi)$ may induce substantial uncertainties in the matter fluxes (Vauchel et al., 2017). Most of this uncertainty is attributable to $C(z_\chi)$, (Gitto et al., 2017), particularly when only a single index sample is taken or when a unique sample position is considered because the relation may change around this position based on the flow conditions.

The index sample representativeness becomes crucial as high-resolution imagery is increasingly used to link remote sensing

reflectance data with suspended sediment concentration (e.g. Mertes et al., 1993; Martinez et al. 2009; 2015, Espinoza-Villar et al., 2012, 2013, 2017; Park and Latrubesse, 2014; Dos Santos et al., 2017). These advanced techniques finally provide a spatially averaged $C(z_\chi)$ value for the finest grain sizes at the water surface of a reach (Pinet et al., 2017), which must be correlated with the total mean concentration transported in the reach of interest (i.e., including the sand fraction when possible) to be a quantitative measurement (Horowitz et al., 2015). Hence, to improve our knowledge of the sediment delivery problem

(Walling, 1983), these empirical relations deserve hydraulic-based undestanding.

In this study, the ratios $\alpha = \langle C \rangle / C(z_\chi)$ measured at 8 gauging stations in the Amazonian foreland were analyzed to identify the main parameters that controlled their variability.

Assuming that the shape of the concentration profiles measured in large Amazonian rivers can be well described using a physically based model for sediment suspension, the possibility of deriving a simple formulation for the ratio $\alpha$ using this

model was investigated.

This assumption is supported by previous studies that specifically showed that the Rouse model (Rouse, 1937) can well describe the suspension of sediments in large tropical rivers (Vanoni, 1979, 1980; Bouchez et al., 2011; Lupker et al., 2011; Armijos et al., 2016). However, the Rouse model predicts a concentration of zero at the water surface, which is where the index concentration is often sampled. To find an alternative, other formulations (Zagustin, 1968; Van Rijn, 1984; Camenen

and Larson, 2008) are compared to the data.

Then, the relevance of the derived model in terms of developing a detailed and reliable sediment flux series with an index method is discussed; specifically, the ability to accurately estimate the model parameters is evaluated. Finally,



recommendations for the optimized collection of index samples from large Amazonian rivers are inferred from the proposed model.

## 2 Materials and methods

### 2.1 Hydrological data acquisition

The hydrological data presented here were collected within the international framework of the Critical Zone Observatory HYBAM (Hydrology of the Amazon Watershed), which is a long-term monitoring program. A Franco-Peruvian team from the IRD (Institut de Recherche pour le Développement) and the SENAMHI (SErvicio NAcional de Meteologia e HIdrologia) operates the 8 gauging stations of the HYBAM hydrological network in Peru; of these, 4 stations control the Andean piedmonts fluxes, and 4 stations control the lowlands (Fig. 1). The three major Peruvian tributaries of the Amazonas (Solimões) River,

i.e., the Ucayali River, the Marañon River and the Napo River, are monitored. The studied sites cover drainage areas ranging from approximately 22,000 km² to 720,000 km² and have mean discharges ranging from 2,100 m$^3$ s$^{-1}$ to 30,300 m$^3$ s$^{-1}$ (Table 1). These large tropical rivers have flows with gradually varied conditions, unimodal and diffusive flood waves (except for the Napo River), and subcritical conditions, which enable backwater effects (Dunne et al., 1998; Trigg et al., 2009).

The Amazonian foreland in Peru has a humid tropical regime (Guyot et al., 2007; Armijos et al., 2013), and large amounts of

runoff are produced during the austral summer. During the austral winter, the maximum continental rainfall is located to the north of the equator, in line with the intertropical convergence zone (Garreaud et al., 2009). Thus, the numerous water supplies from the Ecuadorian sub-basins smooth the seasonality of the Marañon River flow regime. Located further to the south, the Ucayali basin experiences a pronounced dry season (Ronchail and Gallaire, 2006; Garreaud et al., 2009; Lavado et al., 2011; Santini et al., 2014).

The El Niño Southern Oscillation (ENSO) might alter these dynamics, as there are severe low-flow events in El Niño years and heavy rainfall events in La Niña years (Aceituno, 1988; Ronchail et al., 2002; Garreaud et al., 2009). These events affect seriously the sediment routing processes (e.g. Aalto et al., 2003), as other extreme events unrelated to the ENSO (e.g. Molina-Carpio et al., 2017).

### 2.1.1 Sampling strategy

For the reasons outlined in the Introduction, local observers monitor surface index concentrations at each station following a hydrology-based scheme. The sampling depth is typically 20-50 cm below the water surface. The samples are taken in the mainstream and at a fixed position. Additionally, HYBAM routinely uses moderate-resolution MODIS images to determine surface concentrations, and these values are calibrated with in situ radiometric measurements (Espinoza-Villar et al., 2012; Santini et al, 2014; Martinez et al.; 2015).

For calibration purposes (i.e., water level vs discharge and index vs mean concentration), 44 campaigns were conducted during the 2010-2017 period. These campaigns included the collection of 494 discharge measurements, 249 sediment concentration



profiles and 1330 water samples. The dataset covers contrasted regimes, including periods of extreme droughts (e.g. 2010) and periods of extreme floods (e.g. 2012, 2015) (Espinoza et al., 2012, 2013; Marengo et al., 2015). Thus, the sampled concentrations spanned a wide range (Table 1), which was well representative of the river hydrological variability. A 600 kHz Teledyne RDI Workhorse Acoustic Doppler Current Profiler (ADCP) was used and coupled with a 5 Hz GPS sensor to correct

for the movable bed error (e.g. Callède et al., 2000; Vauchel et al., 2017).

A point sampling method was preferred to estimate $\langle C \rangle$ (Filizola, 2003; Guyot et al., 2005; Vauchel et al., 2017) to capture the vertical concentration distribution. The sampling for concentration determination was usually performed at the following height ($h$) from the bed: $\sim 0.98\ h$, $0.75\ h$, $0.5\ h$, $0.25\ h$, sometimes at $\sim 0.15\ h$, and finally at $\sim 0.1\ h$, at three verticals that divided the cross-section according to the river width or the flow rate. Each vertical was assumed to be representative of the

flow in the corresponding sub-section. Sampling was performed from a boat drifting on a streamline immediately after the ADCP measurements were collected. The sampler capacity was 650 mL, with a filling time of ~10 seconds, which allowed for a short time integration along the streamline passing by the sample point. Considering the waves at the free surface, the boat's pitch and roll and the bedforms, the accuracy of the vertical position of the sampler may be evaluated as ± 0.5 m. This variability leads to substantial uncertainty in the zones with high concentration gradients. The operation time was

approximately 2-5 hours, depending on the river sites. Steady conditions were observed during the sampling operation.

### 2.1.2 Analytical methods

The concentrations $C_\phi$ for two main grain size fractions $\phi$ were further determined: the sand fraction ($\phi = s$) was separated from the silt/clay fraction ($\phi = f$) using a 63-µm sieve (cf. Standart Methods ASTM D3977), according to the Wentworth (1922) grain size classification for non-cohesive particles. The water samples were filtered using 0.45-µm cellulose acetate

filters (Millipore) that were then dried at 50°C for 24 hours.

Particle size analysis was performed with a Horiba LA920-V2 laser diffraction sizer. The entire sampled volume was analyzed, with several repetitions demonstrating excellent analytical reproducibility. For each size group $\phi$, the arithmetic mean diameter $d_\phi$ [m] was calculated:

$$d_\phi = \frac{\sum_i d_i X_i}{\sum_i X_i},$$  (3)

where $X_i$ is the relative content in the PSD for the class of diameter $d_\phi$. The settling velocities $w_\phi$ corresponding to the diameters $d_\phi$ derived from the PSD were computed using the Soulsby law (Soulsby, 1997), which assumed a particle density of 2.65 g cm[-3].



## 2.2 Theory for modeling vertical concentration profiles

Schmidt (1925) and O'Brien (1933) proposed a diffusion-convection equation to model the time-averaged vertical concentration distribution $C_\phi(z)$ of grains settling with a velocity $w_\phi$ [m s$^{-1}$]. The grain size, shape and density are considered to be uniform. The equation is expressed as follows:

$$5 \quad \varepsilon_\phi \frac{\partial C_\phi}{\partial z} = -w_\phi \, C_\phi \,, \tag{4}$$

where the term on the left side is the rate of upward concentration diffusion caused by turbulent mixing, balanced by the settling mass flux in the right term. $\varepsilon_\phi$ [m$^2$ s$^{-1}$] is the sediment diffusivity coefficient that characterizes the particle exchange capacity for two eddies positioned on both sides of a horizontal fictitious plane. $\varepsilon_\phi$ is assumed to be proportional to the momentum exchange coefficient $\varepsilon_m$ [m$^2$ s$^{-1}$] (Rouse,1937):

$$10 \quad \frac{\varepsilon_\phi}{\varepsilon_m} = \beta_\phi \,, \tag{5}$$

The $\beta_\phi$ parameter is similar to the inverse of a turbulent Schmidt number (Graf and Cellino, 2002; Camenen and Larson, 2008). It may be depth-averaged (Van Rijn, 1984) or considered to be independent of the height above the bed (Rose and Thorne, 2001).

The main issue of the Schmidt-O'Brien formulation (Eq. (4)) is the expression of the vertical distribution of the sediment mass diffusivity $\varepsilon_\phi$. Once this term is modeled (models are given hereafter), Eq. (4) is depth-integrated from the reference height $z_0$ to the free surface to obtain the expression of the concentration distribution along the water column. The concentration $C_\phi(z_0)$ is then required to determine the magnitude of the profile and can be evaluated using a bedload transport equation (e.g. Van Rijn, 1984; Camenen and Larson, 2008) or measured directly. However, in the case of a sampling operation, the large concentration gradient observed near the riverbed would force the operator to sample water at the reference height $z_0$ with a very high precision to minimize uncertainties; however, achieving such a high level of precision is rarely possible. Hence, it is preferable to choose a more reliable reference concentration in the interval $z = [z_0, h]$. Thus, the following formulae resulting from Eq. (4) are written using $C_\phi(z_\chi)$ instead of $C_\phi(z_0)$ as reference.

Building on Prandtl's concept of mixing length distribution, O'Brien (1933) and Rouse (1937) expressed the sediment diffusion profile with the following parabolic form:

$$25 \quad \varepsilon_\phi(z) = \beta_\phi \, \kappa \, u_* \, z \left(1 - \frac{z}{h}\right), \tag{6}$$

where $\kappa$ is the Von Kármán constant, and $u_*$ [m s$^{-1}$] is the shear velocity.

This expression leads to the classic Rouse equation (Rouse, 1937) for suspended concentration profiles. For $z_\chi \in [z_0, h[$:

$$\frac{C_\phi(z)}{C_\phi(z_\chi)} = \left(\frac{z_\chi}{z} \times \frac{h-z}{h-z_\chi}\right)^{P_\phi}, \tag{7}$$



where $P_\phi = w_\phi / \beta_\phi \kappa u_*$ is the Rouse suspension parameter, i.e., the ratio between the upward turbulence forces and the downward gravity forces. $P_\phi$ is the shape factor for the concentration profile.

The Rouse formulation is widely used in open channels and well suits the observed profiles in the Amazon River (Vanoni, 1979, 1980; Bouchez et al., 2011; Armijos et al., 2016). However, the Rouse formulation predicts a concentration of zero at

the water surface.

Three other simple models for which $C_\phi(h) \neq 0$ have been selected in this work to overcome this problem, i.e., the Zagustin (1968), Van Rijn (1984) and Camenen and Larson (2008) models.

Zagustin (1968) proposed a formulation for the eddy diffusivity distribution based on experimental measurements and a defect law for the velocity distribution.

The following variable changes were introduced: $Z = \sqrt{(h - z)/z}$, and the sediment diffusivity formulation proposed by Zagustin (1968) is:

$$\varepsilon_\phi(Z) = \beta_\phi \, \tfrac{\kappa}{3} \, u_* \, h \, Z \, (1 - Z^2)^3 \,, \tag{8}$$

which leads to an expression with a finite value at the water surface:

$$\begin{cases} \dfrac{C_\phi(z)}{C_\phi(z_\chi)} = \exp\left( P_\phi \left( \Phi(z_\chi) - \Phi(z) \right) \right) \\ \Phi = \dfrac{1}{2} \ln\left( \dfrac{(Z^3+1)(Z-1)^3}{(Z^3-1)(Z+1)^3} \right) + \sqrt{3} \arctan\left( \dfrac{\sqrt{3}\, Z}{Z^2-1} \right) \end{cases}, \tag{9}$$

As the proposed diffusivity profile is slightly different from the parabolic form, this expression leads to $P_\phi$ values that are approximately 7% lower than those obtained with the Rouse theory (Zagustin, 1968).

Van Rijn (1984) proposed a parabolic-constant distribution for sediment diffusivity, i.e., a parabolic profile in the lower half of the flow depth (Eq. (6)) and a constant value in the upper half of the flow depth (Eq. (10)), which corresponds to the maximum diffusivity predicted by the Prandtl-Von Kármán theories. Indeed, some authors have reported measurements with

constant sediment diffusivity in the upper layers (Coleman, 1970; Rose and Thorne, 2001).

$$\varepsilon_\phi(z \geq 0.5\, h) = \tfrac{\beta_\phi}{4} \, \kappa \, u_* \, h \,, \tag{10}$$

For $z \geq 0.5\, h$, the concentration profile is therefore exponential, with a finite value at the free surface:

$$\frac{C_\phi(z)}{C_\phi(z_\chi)} = \left( \frac{z_\chi}{h - z_\chi} \right)^{P_\phi} \exp\left( -4\, P \left( \frac{z}{h} - \frac{1}{2} \right) \right), \tag{11}$$

In addition, Van Rijn (1984) introduced a coefficient to account for the dampening of the fluid turbulence by the sediment

particles. This coefficient value is equal to the unity if the sediment diffusion $\varepsilon_\phi$ distribution is concentration-independent, which was an assumption used in the present work.



Camenen and Larson (2008) showed that the depth-averaged sediment diffusivity $\varepsilon_\phi = \frac{\beta_\phi}{6} \kappa u_* h$ is a reasonable approximation of the Prandtl-Von Kármán parabolic form (Eq. (6)) that does not significantly affect the prediction of the concentration profiles in large rivers ($P_\phi < 1$), except near the boundaries. This simple expression for $\varepsilon_\phi$ lead to an exponential sediment concentration profile:

$$\frac{C_\phi(z)}{C_\phi(z_\chi)} = \exp\left(\frac{6 P_\phi}{h}\left(z_\chi - z\right)\right), \tag{12}$$

This profile has practical interest: there is no need to define the reference level $z_0$ accurately or estimate the corresponding concentration $C_0$ (Camenen and Larson, 2008).

### 2.3 A general expression for the ratio $\alpha$

### 2.3.1 Hypothesis and formalism

$C_\phi(z)$ can be expressed by each of the models presented in this section (Eqs. (7), (9), (12)) and replaced into Eq. (1) to calculate $\langle C_\phi \rangle$. Then, the development of the expression $\alpha_\phi = \langle C_\phi \rangle / C_\phi(z_\chi)$ would lead to the following equation, which is similar to Eq. (2), where the parameters driving $\alpha_\phi$ are identified:

$$\frac{\langle C_\phi \rangle}{C_\phi(z_\chi)} = \alpha_\phi\left(z_0, z_\chi, h, P_\phi(d_\phi, \beta_\phi, u_*), u(h, z_0, u_*)\right), \tag{13}$$

However, the PSDs observed in large rivers are rather broad (e.g. Bouchez et al., 2011; Lupker et al., 2011; Armijos et al., 2016) and may be binned in a range of $n$ grain size fractions $\phi$, as modeling the concentration profiles requires the diameter of sediment in suspension $d_\phi$ to be almost constant throughout the water depth if there is not a narrow PSD.

Assuming that the interaction between sediment classes $\phi$ is negligible, it is possible to apply Eq. (7) and use a multiclass configuration to describe the PSD. Moreover, if the velocity distribution is vertically uniform ($dC_\phi/C_\phi \gg du/u$) (Camenen and Larson, 2008), the multiclass model becomes:

$$\alpha = \sum_{\phi=1}^{n} \alpha_\phi\left(\frac{z_0}{h}, \frac{z_\chi}{h}, P_\phi(d_\phi, \beta_\phi, u_*)\right) X_\phi, \tag{14}$$

where $X_\phi$ is the mass fraction of each grain size fraction in $C_\phi(z_\chi)$ ($\sum_{i=1}^{n} X_\phi = 1$).

A key issue is then to provide a proper model of the PSD using a limited number of sediment classes. In this study, the available dataset provides concentrations for fine particles $0.4 < d_f < 63$ μm) and sand particles ($d_s \geq 63$ μm). Then, the ratio $\alpha$ may be formalized as follows:

$$\alpha = X_f \alpha_f + X_s \alpha_s, \tag{15}$$

Thus, if at height $z_\chi$ the mass fraction of each group is accurately known after sieving, $\alpha$ may be calculated for the whole PSD.





### 2.3.2 Model proposed for the ratio $\alpha_\phi$ prediction in Amazonian large rivers

The depth-integration of the Camenen and Larson formulation (Eq. (12)) is considered a reasonable approximation of the measured $\langle C \rangle$ in large rivers (Camenen and Larson, 2008), with a simple expression that is independent of the $z_0$ term, which differs from the other theories presented above.

Moreover, in the next section, the fit of the suspension models to the measured concentration profiles will show that the Zagustin model provides the best fit to the observations, particularly in the upper layer of the flow.

Thus, in this work, $C_\phi(z_\chi)$ will be expressed with the Zagustin model (Eq. (9)), and $\langle C \rangle$ will be expressed with the Camenen and Larson model.

Because the Zagustin model causes the Rouse number ($P'_\phi$) to be slightly smaller than that calculated with the Rousean model

($P'_\phi \approx 0.93\, P_\phi$, according to Zagustin, 1968), we obtain the following expression for predicting the ratio $\alpha_\phi$:

$$\alpha_\phi(z_\chi, P_\phi) = \frac{\exp\left(6\, P_\phi\, \frac{z_r}{h}\right)\left(1 - \exp(-6\, P_\phi)\right)}{6\, P_\phi \exp\left(0.93\, P_\phi\left(\Phi(z_r) - \Phi(z_\chi)\right)\right)} \quad , \tag{16}$$

where $z_r$ is a reference height required for expressing $C_\phi(z_\chi)$ with the Zagustin model ($z_r$ replaces $z_\chi$ in Eq.(9)). Taking $z_r = 0.5\, h$, the previous expression is simplified:

$$\alpha_\phi(z_\chi, P_\phi) = \frac{\exp(3\, P_\phi)\left(1 - \exp(-6\, P_\phi)\right)}{6\, P_\phi \exp\left(0.93\, P_\phi\left(\Phi\left(\frac{h}{2}\right) - \Phi(z_\chi)\right)\right)} , \tag{17}$$

Nevertheless, other formulations might be inferred from the suspension models. For instance, the Camenen and Larson formulation could be alternatively used to model $C_\phi(z_\chi)$ in the central region of the flow $[0.2\, h, 0.8\, h]$, which leads to a simpler expression:

$$\alpha_\phi(z_\chi, P_\phi) = \frac{1}{6\, P_\phi} \exp\left(6\, P_\phi\, \frac{z_\chi}{h}\right)\left(1 - \exp(-6\, P_\phi)\right), \tag{18}$$

### 2.4 Model fitting strategy

To obtain a reach-scale profile, the fit to the concentrations averaged at each normalized depth $z/h$ was assessed. It was assumed that the energy gradient, the mean bed roughness factor and the mean diameter did not significantly change from one sub-section to another, even if the point-to-point variability was high (Yen, 2002). Thus, the depth becomes the main factor influencing the $P_\phi$ in the transverse direction. In the cross-sections studied here, the variation in depth from one vertical to the next was not sufficient to significantly influence the Rouse number. $C_\phi(z_\chi)$ is then the average of several representative

samples taken across the river width at the same relative height ($z_\chi/h$).

After the first data cleaning of the sampled points, a robust and iteratively re-weighted least squares regression technique was used to minimize the influence of the outlier values. The weight values ($W$) between $z_0$ and $h$ were assigned with the following



parabolic function, similar to the eddy diffusivity expression (Eq. (6)): $W(z) = z\,(1 - z)$. Thus, the half-depth point, where the mixing term is the highest, has the largest influence.

Based on the ADCP velocity profile measurements, the parameter $z_0$ was fixed at $z_0 = 10^{-3}\, h$. Indeed, when $z_0 < 10^{-2}\, h$, $\langle C \rangle$ is no longer sensitive to $z_0$ (Eq. (1)), even if the Rouse number is not accurately known (Van Rijn, 1984). Hence,

$(h - z_0)/(z - z_0) \approx z/h$ can be assumed when considering reach-scale flow conditions.

## 2.5 Shear velocity estimation from ADCP transects

ADCP velocity transects were used to estimate the shear velocities $u_*$ from the vertical velocity gradient through the fit of the logarithmic inner law (e.g. Sime et al, 2007; Gualtieri et al., 2018). Before the fitting procedure, the velocity profiles were averaged over a spanwise length of 60 m around each concentration profile position. Then, an average of the fitted shear

velocities was calculated for each ADCP measurement.

The imprecise knowledge of the exact bed elevation, the side lobe interferences, the beam angle, which induces a large measurement area, and the instrument's pitch and roll all cause the ADCP velocity data to be inaccurate in the inner flow region. However, a fit over the entire height of the measured velocity ($\sim 0.06\, h$ to the ADCP "blanking depth" plus the transducer depth) leads to more robust shear velocity values. For that reason, the shear velocities were assessed in the zone

between $0.1\, h$ and $0.85\, h$.

## 3 Results

### 3.1 Data analysis

#### 3.1.1 Index concentration relations calibrated for surface index samples

The empirical relations for the total estimated concentration (i.e., of fine particles and sands) from the surface index samples

were calibrated (Eq. (2)). The measured $\alpha$ values show there is variability between the sites and among the flow conditions (Fig. 2).

Indeed, if the $\alpha$ ratios are similar ($1.3 < \alpha < 1.5$) at the three stations monitoring the Ucayali basin fluxes (i.e., LAG, PIN, and REQ) and at CHA in the Huallaga basin, different trends with larger scatter are observed in the Marañon basin. The regression slope at REG (2.3) and for the Napo River at BEL (1.7) are higher than the previous ones. The regression slope at

BOR is similar to that at LAG, PIN, REQ and CHA. However, the measured $\alpha$ values fluctuate between two main trends, which were represented by the CHA-LAG-PIN-REQ group and the BEL-REG group. At TAM, similarly, the $\alpha$ values rather follow the REG trend at low concentrations before evolving between the REQ and REG trends.

This variability reflects differences in the basin characteristics (e.g. lithology and climate spatial distributions) and relates to the sediment routing in the floodplain. In the Marañon lowland, the left bank tributaries supply almost 55% of the water

discharge and could significantly contribute to the river sediment load. Indeed, the Napo River example (Laraque et al., 2009;




Armijos et al., 2013) shows that the lowland can be a main sediment source for these Ecuadorian tributaries. This secondary source provides coarser elements than does the central Andean source. As the bed slope and discharge of the lower Marañon are higher than those of Ucayali, the river can more easily route this material to its outlet. Then, the two main trends observed for the $\langle C \rangle / C(z_\chi)$ relations may be roughly interpreted as a central Andean trend (i.e., primary source) and a lowland trend

(i.e., secondary source), with differences in mineralogy and PSD. The BOR and TAM, which are two stations located just after a confluence of rivers influenced by these two sediment sources, adequately show these source mixing dynamics (Fig. 2).

The concentration dataset highlights the control of the sand mass fraction $X_s$ on the ratio $\alpha$ (Fig. 3): $\alpha$ increases with $X_s$. Indeed, the finest particles of the PSD are dominant in the $C_\chi$ sampled in the upper layers of the flow. However, this washload

is supply-limited, depending on the matter availability, rainfall upon the sources, and sediment entrainment processes occurring on the weathered hillslopes. The washload is then routed through the foreland without important mass fluctuations (e.g. Yuill and Gasparini, 2011). Significant exchanges between the floodplain and main channel lead to some dilution but also to some remobilization of the huge floodplain sediment stocks of the coarser elements that were previously deposited (e.g. clay aggregates, silts and fine sands). As the particle size increases, its transport regime may be considered to be capacity-limited,

depending only on the available energy to route the sediments. Thus, this sediment load is gradually decoupled from the washload in the floodplain, and the washload concentration is no longer a good proxy of the coarse particle concentration. The floodplain incision mechanically increases the sand mass fraction $X_s$ in the suspended load. Implicitly, the PSD mean diameter shifts with $X_s$, but it does not mean that there is any change in the physical properties (e.g. diameter, density and shape) of the sand fraction. This shift directly affects $\alpha$, as the vertical concentration gradient depends on the balance between

the turbulence strength and the settling velocity (Eq. (4)). This result highlights the key challenge of providing a proper model of the PSD using a limited number of sediment classes, and valid the discrete approach proposed to model $\alpha$.

### 3.1.2 Particle size profiles

The measured PSD shows a multimodal pattern, similar to that of Fig. 4a. This example of a global PSD that includes the entire particle size range was deconvoluted, assuming a mixture of lognormal sub-distributions (e.g. Masson et al., 2018). On

the left side of the PSD, a weak lognormal mode was detected in the clay range, but it was negligible in comparison to the silt volume. A fairly uniform fraction of fine sands that were transported in suspension throughout the water column with a near constant mode over depth was identified. A coarser material group transported either as a mixed-load or in saltation (corresponding roughly to the diameters up to the 10[th] percentile of the riverbed PSD) appeared as a random factor, which complicated the prediction and measurement of the concentration profiles when the shear stress was large enough to uplift

substantial amounts of these grains in the inner region of the flow.





Concerning the whole $d_f$ dataset (Fig. 4b), no vertical gradient was observed for fine sediments, indicating there was homogeneous mixing throughout the water column, except near the air-water interface, where the calculated $d_f$ tended to decrease.

On the other hand, a small gradient was observed for the sand fraction. Indeed, $d_s$ varied from approximately 300 to 500 μm

near the bottom to 80 - 100 μm near the surface. The increased $d_s$ in the bottom $0.2\,h$ of the water column may be explained by bed material inputs (see yellow distribution in Fig. 4a).

Nevertheless, modeling the PSD with two size groups, which were characterized by a diameter $d_\phi$ that was almost constant throughout the water column, was reasonably suitable for the observed PSD, although two more classes (i.e., clay and bed material) could be considered to improve this model.

Thus, an average of the diameters derived from the PSD was calculated to summarize the PSD data into one single mean diameter $d_\phi$ [m] per site for each size group $\phi$.

### 3.2 Suspension model suitability with the measured profiles

The suspension models (Eqs. 7, 9, 11, 12) were fitted to the concentration data to evaluate their suitability to the observed profiles. The dataset confirmed that $P'_\phi \approx 0.93\,P_\phi$.

The fitted $P_\phi$ values showed low variability and were summarized by single average values per site, as shown in Table 2. This result indicates there is a dynamic equilibrium between $w_\phi(d_\phi)$ and $u_*$ under nominal flow conditions, although some extreme values ($P_\phi > 0.5$) were measured during severe drought events at the lowland stations.

The Rouse numbers obtained for the fine fraction reflect a suspension regime that is close to the ideal washload ($P_f < 0.1, 1 \leq \alpha_f(h, P_f) \leq 1.5$). Additionally, regarding the Rouse numbers corresponding to the sand fraction, they reflect a well-developed

suspension in the entire water column for the piedmont station group ($0.2 < P_s < 0.3$) and for the lowland station group ($0.35 < P_s < 0.45$), with a significant concentration gradient ($2.3 \leq \alpha_s(h, P_s) \leq 7.5$).

Given the availability of a single mean value of $w_\phi(d_\phi)$ per site and size group, single $\beta_\phi = w_\phi/P_\phi \kappa u_*$ corresponding values were calculated considering a mean shear velocity per site (Table 2).

### 3.2.1 Sediment diffusivity profiles

The diffusivity profiles $\varepsilon_\phi(z)$ were derived from the measured concentration profiles with the discrete form of Eq. (4). This calculation requires accurate concentration and sampling height values, particularly for the fine fraction, which experiences low vertical concentration gradients.

Nevertheless, the overall shapes of the derived $\varepsilon_\phi(z)$ profiles were in good agreement with the Rouse and Zagustin theories and were slightly closer to the second one (Fig. 5). Given the high scatter of the diffusivity values, Camenen and Larson's

expression of depth-averaged diffusivity is a reasonable approximation, except near the bottom and top edges of the diffusivity



profiles, where the data departs gradually from this model. However, the constant diffusivity value suggested by Van Rijn (1984) for the upper half of the water column clearly overestimates the diffusivity for $z > 0.75\,h$.

The diffusivity around $z = 0.75\,h$ is, however, overestimated by all the models. The low concentrations near the water surface could result in an underestimation of the $\varepsilon_\phi(z \approx 0.75\,h)$ values calculated from the difference $\Delta C_\phi = C_\phi(z \approx 0.5\,h) -$

$C_\phi(z \approx h)$ (Eq. (4)). Thus, detailed measurements are required in the upper layer of the flow to confirm the shapes of the $\varepsilon_\phi(z)$ profiles in this zone where the water-air interface and the secondary currents can influence the turbulent mixing profiles.

### 3.2.2 Concentration profile suitability

Overall, the suspension models (Eqs. 7, 9, 11, 12) fit the measured concentration profiles with only a small relative error (Fig. 7), except near the edges where the highest discrepancies between the two exponential expressions (Van Rijn, 1984; Camenen

and Larson, 2008) and the Rouse and Zagustin models appear.

The concentrations sampled at the bottom edge confirmed the general shape of the Zagustin and Rouse models, despite the uncertainties in the concentrations measured in this zone.

Near the water surface, the non-zero values predicted by the Zagustin model were often the closest to the observed concentrations.

The use of the Camenen and Larson model to calculate $\langle C_\phi \rangle$ seems to be a reasonable approximation. Indeed, for the range of nominal Rouse numbers considered here ($P_\phi < 0.6$), the bottom concentration gradient has little influence on $\langle C_\phi \rangle$ because the velocity drops to zero in this zone. Moreover, the top-layer concentrations is too low to weight significantly on $\langle C_\phi \rangle$.

The comparison between the predicted and observed mean $\alpha_\phi$ values per site and size group (Fig. 7a) allows for the validation of the general model proposed in this work (Eq. (17)). To show the model's ability to predict how $\alpha_\phi$ changes with flow

conditions at one specific site, this model was also compared with all the $\alpha_\phi$ values measured at the water surface and at mid-depth (Fig. 7a). The observations follow the model trend well, despite the high scatter of the $\alpha_s(h, P_s)$ values, which is caused by the low diffusivity and concentration in coarse material in this zone and by the uncertainty of the exact z-position of the samples. At mid-depth, the $\alpha_s(0.5\,h, P_s)$ values has lower scatter.

Nevertheless, the $\alpha_s$ sensitivity to the Rouse number remains moderate for most of the hydraulic conditions encountered,

except for the extremely low waters, i.e., when $P_s > 0.5$. The $\alpha_f$ sensitivity to changes in flow conditions is very low. Then, considering the poor contribution of the low waters to the sediment budget and the small Rouse number variations for the nominal hydraulic conditions at a specific site (Table 2), the use of the mean $\alpha_\phi$ coefficients per site seems to be reasonable for assessing reliable sediment budgets.

Regarding the simplified model (Eq. (18)), a reasonable approximation is expected to be found in the central region of the

flow, but the values gradually depart from the observations near the water surface and the riverbed.





Finally, the mean ratios $\alpha(h)$ per site were computed (Eq. (15)) using the predicted mean $\alpha_f(h, P_f)$ and $\alpha_s(h, P_s)$ (Eq. (17)) and the mean mass fractions $X_f$ and $X_s$ measured at the water surface (Table 2). The observed vs. predicted $\alpha$ ratios are in excellent agreement ($r^2 = 0.97$) (Fig. 7b) and valid the prediction ability of the model when the Rouse numbers are accurately known.

## 4 Discussion on the model applicability

The equations proposed in this work (Eqs. 17, 18) for modeling the $\alpha_\phi(z_\chi, P_\phi)$ are highly sensitive when the Rouse number is fairly large ($P_\phi > 0.4$) and when $z_\chi \approx h$ (Fig. 7a). Therefore, estimating $P_\phi$ with a low uncertainty is a key issue to predict accurate $\alpha_\phi$ ratios during sediment concentration monitoring.

This estimation can be achieved (1) through the estimation of the hydraulic parameter $u_*$, $w_\phi$ and $\beta_\phi$, or (2) empirically using detailed point concentration measurements. Then, if the Rouse number variability is significant during the hydrological cycle, the empirical relationship between $u_*$ or $h$ and the measured $P_\phi$ may be calibrated.

### 4.1 Estimation of the diffusivity ratio $\beta_\phi$

For many decades, studies based on flume experiments or measurements in natural rivers have shown that $\beta_\phi$ usually departs from the unity. The sediment diffusivity increases ($\beta_\phi > 1$) with bedforms or movable bed configurations (Graf and Cellino, 2002); specifically, the boundary layer thickness tends to be thin just before the bedforms crest, and then it peels off at the leeward side (Engelund and Hansen, 1967; Bartholdy et al., 2010). This trend implies there are anisotropic macro-turbulent structures, with eddies that convect large amounts of sediments to the upper layers and settle further after eddy dissipation. Thus, bedforms locally modify the ratio between the laminar and turbulent stresses, inducing different lifting profile shapes in the inner region (e.g. Kazemi et al., 2017) and causing the mixing length theory to fail in the overlap region. Centrifugal forces driven by turbulent motion and applied on the grains could also enhance the particle exchange rate between eddies (Van Rijn, 1984).

Conversely, the suspension is dampened ($\beta_\phi < 1$) when the large suspended particles do not fully respond to all velocity fluctuations, such as passive scalars.

Van Rijn (1984), Rose and Thorne (2001) and Camenen and Larson (2008) attempted to model $\beta_\phi$ as a function of the ratio $w_\phi/u_*$ for sand and silt particles. However, the measured $\beta_\phi$ encompasses poorly understood physico-chemical processes as well as uncertainties and bias of the $w_\phi$ and $u_*$ estimations, which might partly explain the shifts along the $w_\phi/u_*$ axis between the three pre-cited laws and the $\beta_\phi$ measured in this study (Fig. 8a).





With regard to $w_\phi$, a major difficulty comes from the need to divide the PSD into various size groups and to summarize each sub-distribution with a single characteristic diameter (e.g. mode, median, mean), and different values of $w_\phi(d_\phi)$ are calculated according to the choices made.

The aggregation process is a supplementary complicating factor (Bouchez et al., 2011) but is probably not the main issue in these white rivers with little organic matter (Moquet et al., 2011; Martinez et al., 2015). Indeed, the results of Bouchez et al. (2011) are probably biased because the authors used a single diameter to summarize the entire PSD, which is highly sensitive to the flow conditions. However, this bias would not concern the sand group because the shear modulus experienced in large Amazonian rivers would prevent the formation of large aggregates.

The choice of a settling law (e.g. Stokes; Zanke, 1977; Cheng, 1997; Soulsby, 1997; Ahrens, 2000; Jiménez and Madsen, 2003; Camenen, 2007) may also induce bias on $w_\phi$. In these laws, the sediment density is a key parameter that is often neglected, as natural rivers comprise a diversity of minerals with contrasting density ranges.

On the other hand, the shear velocity estimation also suffers from uncertainties in terms of the velocity measurements and biases that are induced by the method used (Sime et al, 2007).

For instance, the departures from logarithmic velocity profiles increase with the distance to the bed (e.g. Guo et Julien, 2008) in sediment-laden flows (e.g. Castro-Orgaz et al., 2012), which could be relevant to deep Amazonian rivers. Indeed, the mixing length expansion could reach a maximum before the water surface, as the energetic eddy size cannot expand *ad infinitum* far from the flow zone under the influence of bed roughness because of the increasing entropy. The log-law assumptions (i.e., constant shear velocity throughout the water column and mixing length approximation) would no longer be valid, and the velocity profiles would follow a defect law in the outer region. This raises the need to find a suitable model for the velocity distribution in large rivers, leading to an unbiased estimate of the shear velocity.

It is not surprising to find discrepancies between the empirical laws and the observations based on the experimental conditions. Here, the Rose and Thorne (2001) empirical law is the closest to the measured $\beta_\phi$ (Fig. 8a), with departures that seem to be a function of the water level.

We assume that a global correction of the different bias on the $w_\phi/u_*$ term would depend on the flow depth as well as on the skin roughness, which partly influences the formation and expansion of the turbulence structures and thus influences the velocity distribution (Gaudio et al., 2010). Here, $d_s$ is considered instead of the skin roughness height, as few riverbed PSDs are available and because it is a key parameter for the settling law. Thus, the following modification of the Rose and Thorne (2001) law is proposed:

$$\beta_\phi = 3.1 \exp\left(-0.19 \, \frac{u_*\left(\frac{h}{d_s}\right)^{0.6}}{1000 \times w_\phi}\right) + 0.16 \,, \tag{19}$$

This unidimensional law, which extends below the range of $w_\phi/u_*$ usually considered in previous studies, allows for an enhanced prediction of $\beta_s(\pm 0.03)$ and predicts a near constant $\beta_f \approx 0.16$ (Fig. 8a).





Applying Eq. (19) to predict the mean $\beta_\phi$ values per site, the predicted and fitted $P_\phi$ are in good agreement (Fig. 9a), with little scatter when considering the uncertainties in the measured concentrations and therefore on the fitted $P_\phi$.

This result shows that the shear velocity, rather than variations in particle size, mainly controls the Rouse number variability

in a given site (Fig. 9b), which was considered as a constant for all the hydrologic conditions in the current study. The shear velocity is itself driven by the high amplitude of the water depth in Amazonian rivers (Fig. 9c), and it has hysteresis effects at the gauging stations located in the floodplain, which are attributed to the backwater slope variability in these subcritical flood wave contexts (Trigg et al., 2009).

Hence, the accurate monitoring of the water level and knowledge of the river surface slope, even if limited or biased, would

allow for an acceptable prediction of the Rouse numbers, which could be used to establish a single $\beta_\phi$ value per site.

## 4.2 Predicting $d_s$ from the riverbed PSD

For fine particles, $d_f$ can be accurately measured in the water column because the fine particles are well mixed in the flow. Regarding the sand particles, such measurements induce uncertainties due to the particle fluctuations in the current and because the eddy structure development in the bottom layers of the flow swiftly causes strong grain size sorting (Fig. 4). The suspended

sediment particles are thus considerably smaller than the bed load or riverbed particles (Van Rijn, 1984).

The diameter of the suspended sand can be assessed by taking a representative percentile of the riverbed PSD (e.g. Rose and Thorne, 2001). Alternatively, an empirical expression that considers the flow conditions was proposed by Van Rijn (1984).

Here, the Camenen and Larson (2005) formulae for the estimation of reference concentration $C_\phi(z_0)$ was applied in a multiclass way to the riverbed PSD, and it was assumed that the size fractions did not influence each other and there was a

uniform sediment density for all grain sizes (2.65 g cm⁻³). In this formulation, $C_\phi(z_0)$ is a function of the dimensionless grain size $d_*$, the local Shields parameter $\theta_\phi$ and of the critical Shields parameter $\theta_{cr}$ for the inception of transport (Camenen et al., 2014):

$$C_\phi(z_0) = \frac{0.0015\,\theta_\phi}{\exp\left(0.2\,d_* + 4.5\,\frac{\theta_{cr}}{\theta_\phi}\right)},$$  (20)

This first PSD predicted at the transition level $z_0$ is further diffused vertically with the Zagustin model (Eq. (9)), considering

the Soulsby (1997) settling law in the $P_\phi$ calculations. The model underestimates the measured $d_s$ by approximately 10%. This slight discrepancy might be explained by stochastic and ephemeral inputs of coarse bed material in the water column, which are not addressed by the suspension theory.



### 4.3 Sensitivity analysis and recommendations for optimized sampling procedures

The approximation error $\Delta P_\phi$ can be evaluated at $\pm 0.03$ (from Eq. (19), Fig. 9 or Table 2) and propagated to the corresponding $\alpha_\phi(z_\chi, P_\phi \pm \Delta P_\phi)$ (Fig. 11a). The error on $z_\chi$ is not considered here but would increase the $\alpha_\phi$ sensitivity in the zones with a high concentration gradient.

Overall, the relative error on $\alpha_\phi$ remains moderate for all the flow conditions experienced by the rivers studied here (i.e., below $\pm 10\%$ in the central zone of the flow and below $\pm 20\%$ at the water surface), except near the riverbed (Fig. 11a). Nevertheless, for operational applications, this result must be weighted by the relative error profile of the measured index concentrations $\Delta C_\phi(z_\chi) / C_\phi(z_\chi)$.

By substituting $C_\phi(z_\chi)$ with the Zagustin model (Eq. (9)) and assuming $\Delta C_\phi(0.5\ h)/C_\phi(0.5\ h) = \pm 10\%$, it is possible to

model this profile of concentration uncertainty (Fig. 11b) and to derive the relative uncertainty of $\langle C_\phi \rangle$ according to the sampling height under various flow conditions (Fig. 11c).

Here, the considered uncertainty is a simple function of the concentration. However, coarse particles are more sensitive to current fluctuations than are fine sediments. Thus, the sand concentration uncertainty is underestimated, at least in the region of the flow under bed influence $\sim [z_0, 0.2\ h]$, where stochastic uplifts of bed sediments impose high variability on the

concentration.

Furthermore, the sampling frequency as well as the number of index samples taken and their positions are important parameters to consider. The section geometry, the velocity distribution and the transversal movable bed velocity pattern are important guidelines in the selection of a sampling position(s). The integration of the lateral variability of the concentration is not discussed here.

Nevertheless, when considering these assumptions, optimized sampling heights may be defined:

- For fine sediments ( $P_f < 0.1$ ), the most accurate $\langle C_f \rangle$ is obtained when sampling the water column at approximately $0.5\ h$. The sampling can also be achieved at the water surface with a good estimation of $\langle C_f \rangle$ ($\pm 15\%$).

- For the sand fraction at the piedmont stations ($P_s < 0.3$), sampling in the $[0.2\ h, 0.8\ h]$ region is recommended to keep the errors of $\langle C_s \rangle$ below $\pm 20\%$. A sampling at the water surface is still possible, but there will be uncertainties

between $\pm 20$-$40\%$ for $\langle C_s \rangle$.

- For enhanced monitoring of the sand concentration at the lowland stations ($P_s > 0.3$), the $[0.2\ h, 0.8\ h]$ zone is preferred over the water surface, where the $\alpha_\phi$ prediction would require very accurate estimations of $P_s$ and $C_s(h)$.

The proposed $\alpha_\phi$ models (Eqs. 17, 18) allow for achieving a routine protocol with sampling in the central zone of the flow: the $\alpha_\phi$ can be predicted at each sampling time-step, when the section geometry is known and when the flow is sufficiently

stable to estimate $z_\chi/h$. For instance, a single fixed sampling depth could be used.

Alternatively, the Rouse number can be estimated at each sampling time-step from Eq. (7) by sampling two heights $z_{\chi_1}$ and $z_{\chi_2}$ of the water column at each measurement time-step:



$$P_\phi = \frac{\ln\left(\frac{C_\phi(z_{\chi_1})}{C_\phi(z_{\chi_2})}\right)}{\ln\left(\frac{z_{\chi_2}\,(h-z_{\chi_1})}{z_{\chi_1}\,(h-z_{\chi_2})}\right)}, \tag{21}$$

For instance, the concentration at $z_{\chi_1} = 0.7\,h$ and the concentration at $z_{\chi_2} = 0.3\,h$ result in $P_\phi = 0.59\ln\left(C_\phi(0.3\,h)/C_\phi(0.7\,h)\right)$.

When considering nominal flow conditions ($P_\phi < 0.6$), the sampling height above the riverbed $h/e \approx 0.37\,h$ ($e$ being the Euler number) appears to be pertinent for simplified operations, as the ratios of $\alpha_\phi(h/e, P_\phi)$ remain interestingly close to unity ($\pm 10\%$) (Fig. 7a). Thus, $\{\beta_f, \beta_s\} \approx \{0.16, 1\}$ could be simply assumed without inducing large errors in the $\alpha_\phi(h/e, P_\phi)$ estimations. The particles are usually present in a significant amount, and the turbulent mixing is intense (Eq. (6)) while the concentration gradients are moderate, which also allow for more uncertainty regarding $z_\chi$. Interestingly, when considering the depth-averaged velocity $\langle u \rangle \approx u(h/e)$, the sediment discharge on a vertical $q_{ss}$ [g s$^{-1}$ m$^{-2}$] may be expressed as follows:

$$q_{s\phi} \approx (\alpha_\phi = 1 \pm 10\%) \times C_\phi\left(\frac{h}{e}\right) \times u\left(\frac{h}{e}\right), \tag{22}$$

Finally, if sampling in the central zone of the flow is not technically feasible during concentration monitoring, the mean concentration of fine particles may be estimated with surface index sampling or remote sensing (Martinez et al., 2015; Pinet et al., 2017). Then, the sand concentration could be assessed with a sediment transport model that is suitable for large rivers (e.g. Molinas and Wu, 2001; Camenen and Larson, 2008). To parameterize such models, improved space-borne altimeters (e.g. the SWOT mission) and hydrological models are already serious alternatives to in situ discharge, water level and slope measurements (e.g. De Paiva et al., 2013; Paris et al., 2016).

## 5 Conclusion and perspectives

The use of measured concentration profiles with physically-based models describing the suspension of grains in turbulent flow has shown the possibility to derive a simple model for the prediction of $\alpha_\phi$, i.e., for a given particle size group $\phi$. A proper modeling of the PSD using two hydraulically consistent size groups (i.e., fine particles and sand) is first required to obtain a characteristic diameter that is mostly constant for each size group during the hydrological cycle.

The Zagustin profile, with finite values at the water surface, demonstrated the best suitability in relation to the observed data. Nevertheless, the Camenen and Larson model was in good agreement with the observations in the central zone of the flow and was a reasonable approximation of the depth-averaged concentration.

The Rouse number is the main parameter for the $\alpha_\phi$ modeling. Variations in $P_\phi$ during the hydrological cycle may be monitored from a few point concentration measurements or through the calibration of a relation between the $u_*$ or $h$ and the measured $P_\phi$. Alternatively, a function of $w_\phi/u_*$ and $h/d_s$ was proposed to compute $\beta_\phi$ and predict $P_\phi \pm 0.03$.



The sensitivity of the $\alpha_\phi$ model decreases from the boundaries to a zone between $[0.2\ h, 0.5\ h]$, which is based on the flow conditions. At the water surface, the model becomes inaccurate when $P_s > 0.3$, i.e., for flow conditions corresponding to sand suspension in the lowland. In such a context, sampling in the central zone of the flow is preferable for sand concentration monitoring. A pertinent sampling height for optimized concentration monitoring appears to be $z_\chi = 0.37\ h$.

This insight into the hydraulic theory leads to enhanced sediment monitoring practices that can more accurate estimate sediment loads, especially in regions with limited available data, such as the Amazonian basin. Indeed, the proposed model is a tool that can be used to predict the ratios of $\alpha_\phi$ and $\alpha$ and can also be used to select a proper sampling height for optimized monitoring. Extensively, the model allows for detailed uncertainty analysis on the $\langle C \rangle$ derived from an index method. Finally, where the cross-section geometry is well known, the model could allow for an accurate estimation of $\langle C_f \rangle$ with surface

concentration monitoring by remote sensing. Coupling this monitoring with a sand transport model suitable for large rivers could ensure a better understanding of the sediment dynamics in the Amazonian basin ad can be combined with the use of gauging stations without in situ concentration data.

**Data availability**

The data that support the findings of this study are available from the following data-repository (Santini et al., 2018):
https://doi.org/10.6096/DV/CBUWTR. Extra data (water levels, suspended concentration time series…) are also available from the corresponding author upon request and on the CZO HYBAM website: www.so-hybam.org.

**Author contributions**

-    Conceptualization, data analysis, investigation, methodology and codes development: WS, BC, JLC, PV
-    Manuscript preparation and data discussion; WS, BC, JLC, JMM, JLG
-    Funding acquisition, project administration, supervision: JMM, JLG, WS, WL, MAP
-    Hydrologic data acquisition: WS, PV, JC, JJPA, REV, JLG
-    Laboratory analysis: NA, FJ, WS

**Competing interests.** The authors declare that they have no conflict of interest.

**Acknowledgements**

This work has been funded by the French National Research Institute for Sustainable Development (IRD), the National Center for Scientific Research (CNRS - INSU) and the Peruvian Hydrologic and Meteorology Service (SENAMHI). The authors



would especially like to acknowledge all their colleagues of the National Agrarian University of La Molina (UNALM) and of the Functional Ecology and Environment laboratory (EcoLab) who contributed to the analysis of the data used in this study.

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

| | | | | | | | | Suspended sediment concentration ranges (mg L$^{-1}$) | | Particle size distribution | |
|---|---|---|---|---|---|---|---|---|---|---|---|
| Station code | Site | River | Basin area (km²) | Mean discharge (m³ s$^{-1}$) | Period considered | Number of samples | Number of concentration profiles | Clay & silt | Sands | Number of PSD profiles | Number of PSD samples |
| LAG | Lagarto | Ucayali | 191000 | 6700 | 2011-2017 | 142 | 27 | 9 - 1340 | 1 - 3700 | 2 | 10 |
| PIN | Puerto Inca | Pachitea | 22000 | 2100 | 2012-2015 | 105 | 27 | 180 - 1600 | 6 - 2800 | 0 | 1 |
| REQ | Requena | Ucayali | 347000 | 12100 | 2010-2015 | 213 | 36 | 110 - 1600 | 5 - 2300 | 4 | 25 |
| BOR | Borja | Marañon | 115000 | 5200 | 2010-2015 | 130 | 27 | 40 - 1250 | 2 - 3400 | 2 | 8 |
| CHA | Chazuta | Huallaga | 69000 | 3200 | 2010-2015 | 141 | 27 | 60 - 1450 | 5 - 2330 | 0 | 0 |
| REG | San Regis | Marañon | 362000 | 18000 | 2010-2015 | 226 | 39 | 60 - 600 | 5 - 1600 | 3 | 16 |
| TAM | Tamshiyacu | Amazonas | 720000 | 30300 | 2010-2015 | 223 | 39 | 60 - 960 | 1 - 1600 | 2 | 21 |
| BEL | Bellavista | Napo | 100000 | 7400 | 2010-2015 | 150 | 27 | 40 - 340 | 4 - 830 | 1 | 7 |
| Table summary | 8 | Amazonas | 820000 | 37700 | 2010-2017 | 1330 | 249 | 9 - 1600 | 1 - 3700 | 14 | 88 |

**Table 1:** Hydrologic and sample dataset for the 8 sampling stations.

| | | | | | Mean results for the fine particle fraction | | | | | | Mean results for the sand fraction | | | | |
|---|---|---|---|---|---|---|---|---|---|---|---|---|---|---|---|
| Station code | $R_h$ (m) | $u_*$ (m s$^{-1}$) | $X_s$ $z_{index}=h$ | $d_f$ (µm) | $w_f$ (m s$^{-1}$) | $P_f$ | $\beta_f$ | $\alpha_f$ $z_{index}=h$ | $\alpha_f$ $z_{index}=0.5h$ | $d_s$ (µm) | $w_s$ (m s$^{-1}$) | $P_S$ | $\beta_s$ | $\alpha_s$ $z_{index}=h$ | $\alpha_s$ $z_{index}=0.5h$ |
| LAG | 6.8 | 0.10 | 15% | 16 ± 3% | 2.5E-04 | 0.03 ± 16% | 0.21 | 1.1 | 1.0 | 148 ± 04% | 1.8E-02 | 0.27 ± 09% | 1.7 | 3.5 | 1.1 |
| PIN | 7.2 | 0.14 | 20% | 14 ± 3% | 1.9E-04 | 0.01 ± 14% | 0.24 | 1.0 | 1.0 | 124 ± 02% | 1.4E-02 | 0.23 ± 12% | 1.1 | 2.3 | 1.0 |
| REQ | 12.5 | 0.09 | 7% | 16 ± 4% | 2.5E-04 | 0.05 ± 08% | 0.14 | 1.2 | 1.0 | 114 ± 07% | 1.2E-02 | 0.39 ± 07% | 0.8 | 6.4 | 1.2 |
| BOR | 7.9 | 0.17 | 10% | 16 ± 4% | 2.5E-04 | 0.02 ± 17% | 0.17 | 1.0 | 1.0 | 118 ± 10% | 1.3E-02 | 0.28 ± 16% | 0.6 | 4.3 | 1.1 |
| CHA | 7.4 | 0.20 | 16% | | | 0.01 ± 19% | | 1.1 | 1.0 | | | 0.24 ± 07% | | 2.6 | 1.0 |
| REG | 16.7 | 0.14 | 14% | 15 ± 3% | 2.2E-04 | 0.07 ± 10% | 0.06 | 1.4 | 1.0 | 132 ± 07% | 1.4E-02 | 0.44 ± 05% | 0.6 | 7.5 | 1.2 |
| TAM | 18.6 | 0.12 | 6% | 17 ± 4% | 2.8E-04 | 0.08 ± 14% | 0.07 | 1.5 | 1.0 | 141 ± 10% | 1.7E-02 | 0.44 ± 05% | 0.8 | 8.0 | 1.2 |
| BEL | 10.1 | 0.10 | 17% | 19 ± 2% | 2.5E-04 | 0.04 ± 16% | 0.15 | 1.2 | 1.0 | 192 ± 19% | 2.8E-02 | 0.32 ± 08% | 2.1 | 4.3 | 1.2 |
| Mean | 10.9 | 0.13 | 13% | 16 | 2.4E-04 | 0.04 | 0.16 | 1.2 | 1.0 | 138 | 1.6E-02 | 0.33 | 1.1 | 4.8 | 1.1 |
| Method | Measured | Fitted on u(z) (Log-law) | Measured | Derived from measured PSD | Soulsby law | Fitted on C(z) (Rouse model) | $w_f/(P_f \kappa u_*)$ | Fitted on obs. | Fitted on obs. | Derived from measured PSD | Soulsby law | Fitted on C(z) (Rouse model) | $w_S/(P_S \kappa u_*)$ | Fitted on obs. | Fitted on obs. |

**Table 2:** Summary of suspended sediment transport parameters for each site and size group.



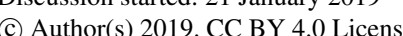

**Figure 1:** Location of the sampling sites in the Amazonian basin. Blue squares: Piedmont gauging stations; yellow dots: lowland gauging stations. In cyan: flooded areas.




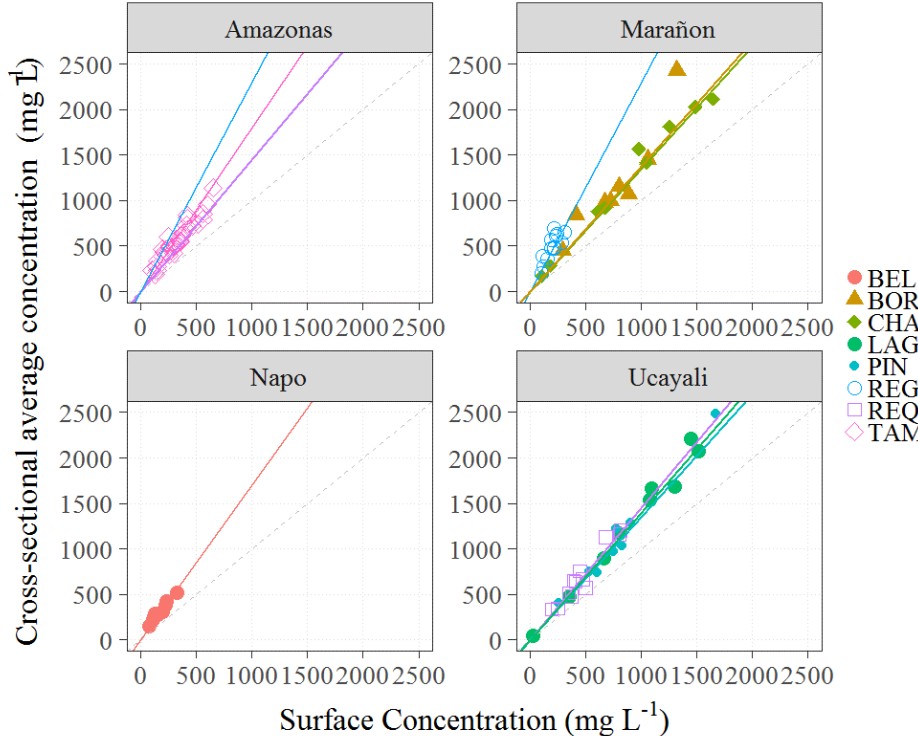

**Figure 2:** Measured $\alpha = \langle C \rangle / C(h)$ ratios with fitted lines, staked by river basin. For the Amazonas River basin at TAM, the REG and REQ trend lines were reported. Dashed lines: first bisector.

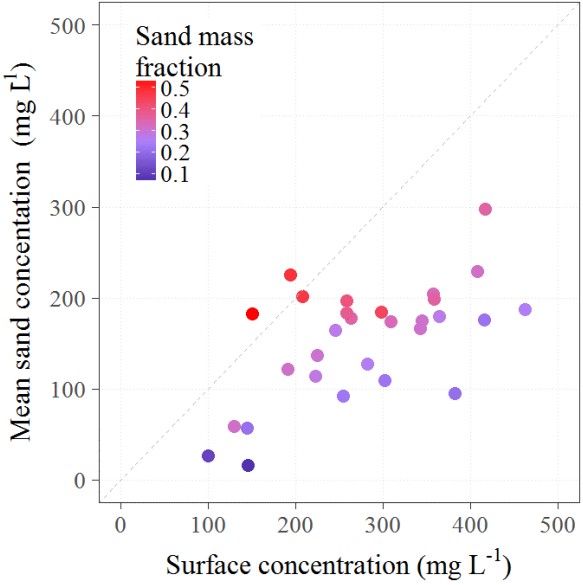

**Figure 3:** Mean sand concentration vs surface concentration at TAM. Points are colored according to the sand mass fraction of the total concentration.







**Figure 4:** (a) Multimodal modeling of a typical PSD vertical profile. Gray lines: PSD measured at the Requena gauge station (03-16-2015) on the Ucayali River. Sampling depths are mentioned in the subtitles. Green, blue, pink and yellow correspond roughly to the following particle size groups: clays, silts and flocculi, very fine sands – fine sands, and bed material, respectively. The dashed gray line is the sum of the sub-distributions. (b) Particle diameters $d_\phi$ measured at the 8 sampling stations for the fine and sand fractions.





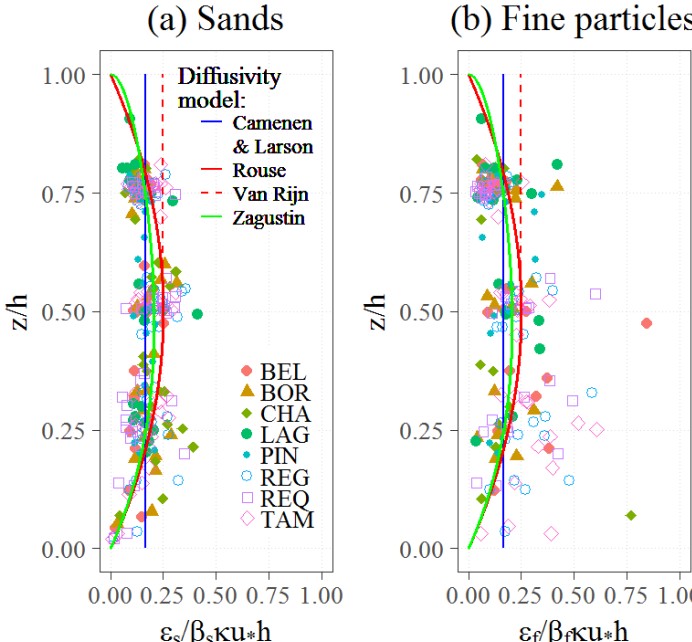

**Figure 5:** Sediment diffusivity values derived from the measured concentration profiles.

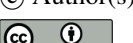





**Figure 6:** Typical examples of measured concentration profiles $C_\phi(z/h)$, fitted with the Rouse, Van Rijn, Zagustin and Camenen and Larson models.





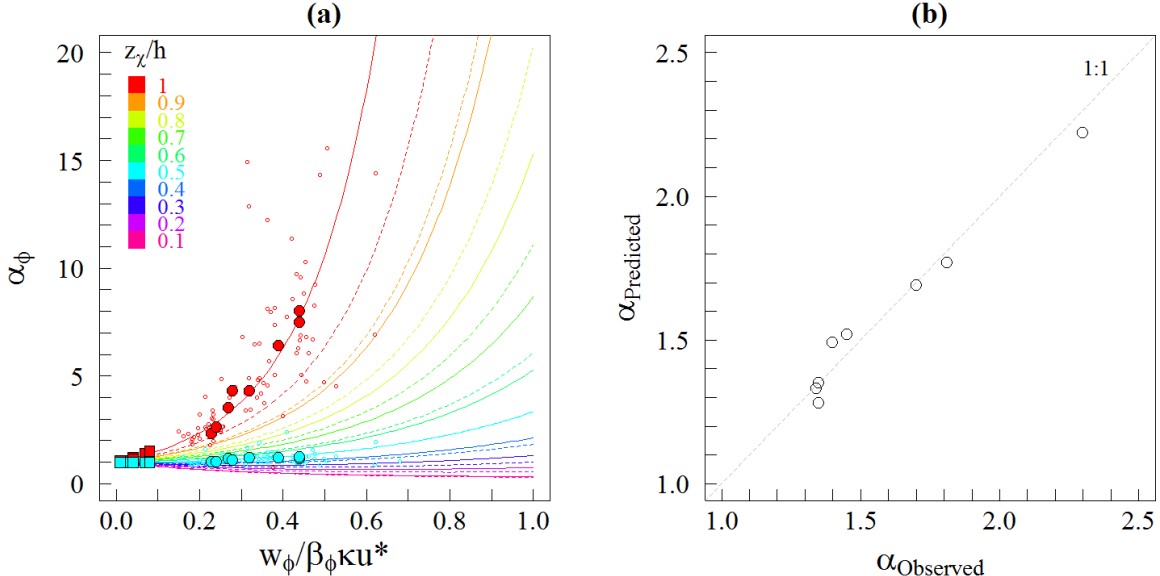

**Figure 7:** (a) Predicted and observed $\alpha^{\phi_i}$ ratios as a function of the Rouse number. Filled circles and squares are the mean $\alpha_\phi$ values observed per site for the sand and fine mass fractions, respectively. Unfilled circles: measured $\alpha_s$ values. The red to pink rainbow set of solid lines correspond to the general model prediction (Eq. 17), and each represents 10% of the water height. Dashed lines are for the simplified model (Eq. 18). (b) Predicted vs. observed mean $\alpha$ ratios per site (i.e., total concentration).

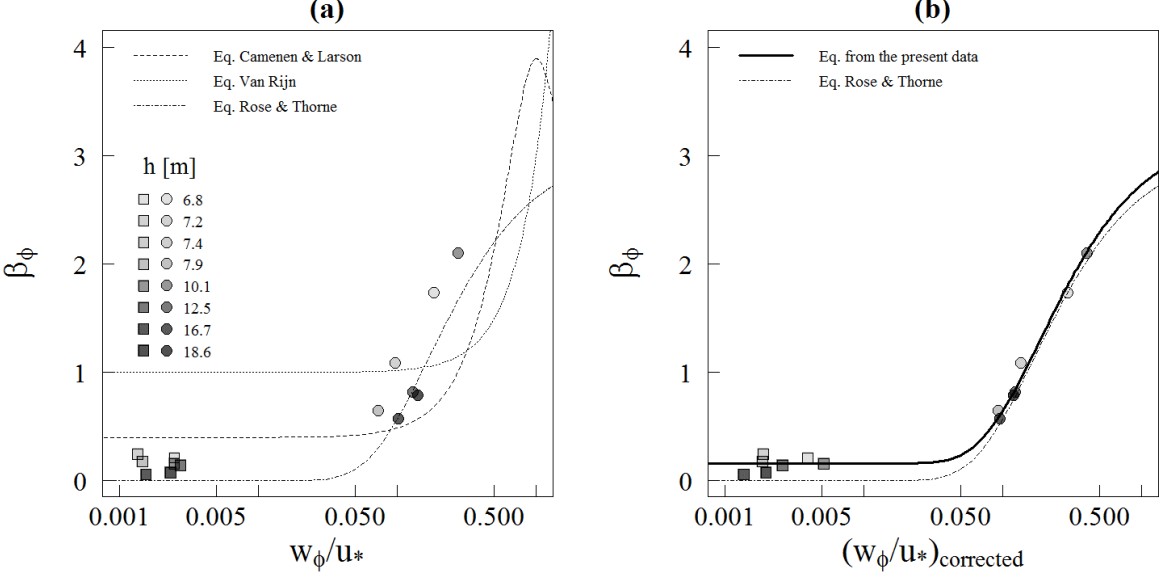

**Figure 8:** (a) $\beta_\phi$ factor as a function of the ratio $w_\phi/u_*$. (b) Idem, after correction of the ratio $(w_\phi/u_*)$. Circles and squares are the mean values of $\beta_s$ and $\beta_f$ calculated per site, respectively.





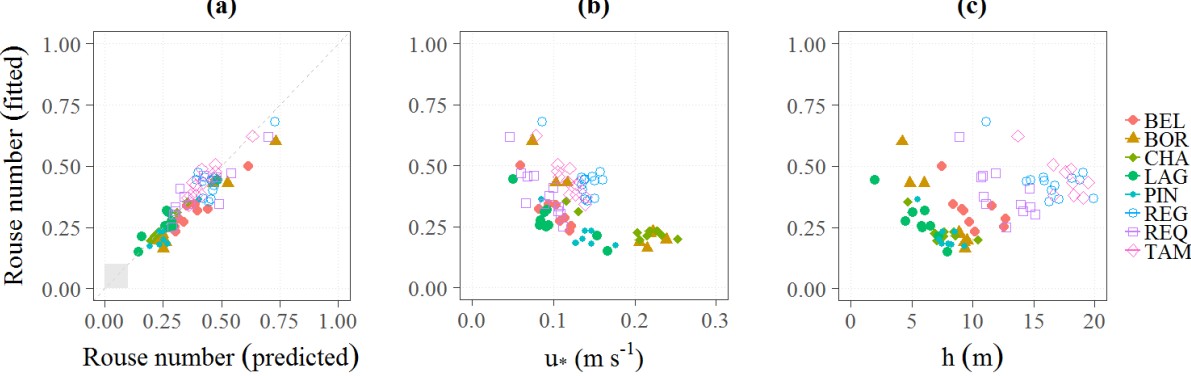

**Figure 9:** Fitted Rouse numbers against: (a) Predicted $P_s$ (the gray square in the bottom-left represents the range of variation of $P_f$) (b) Shear velocities (c) Water levels.

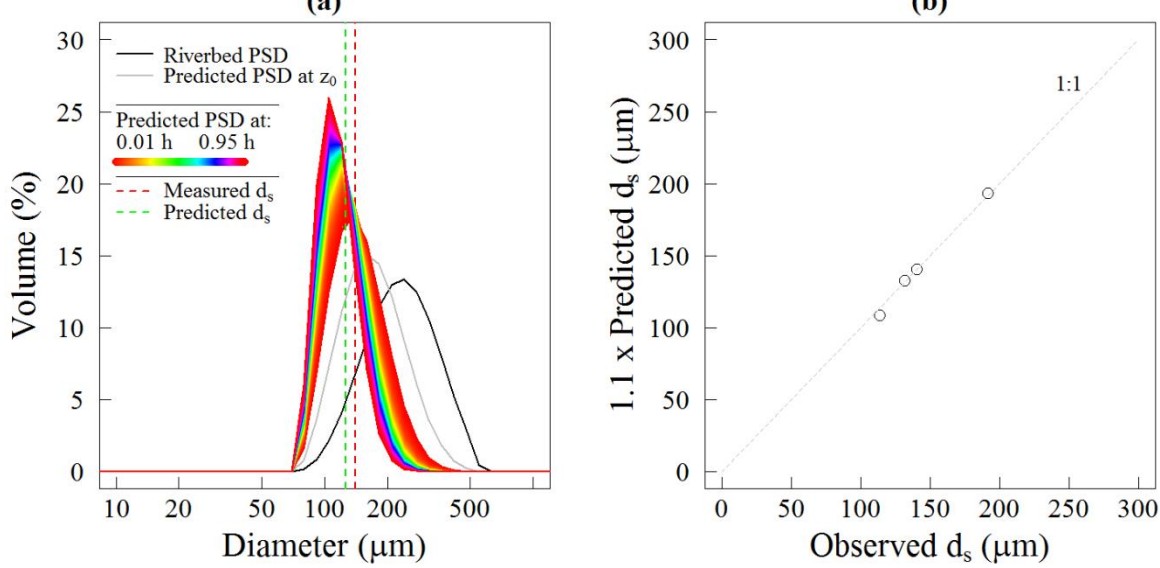

**Figure 10:** (a) Prediction of the mean diameter $d_s$ at the TAM gauging station for $u_* = 0.12$ m s$^{-1}$ (mean flow conditions) (b) Predicted vs measured $d_s$ at BEL, REQ, REG and TAM.





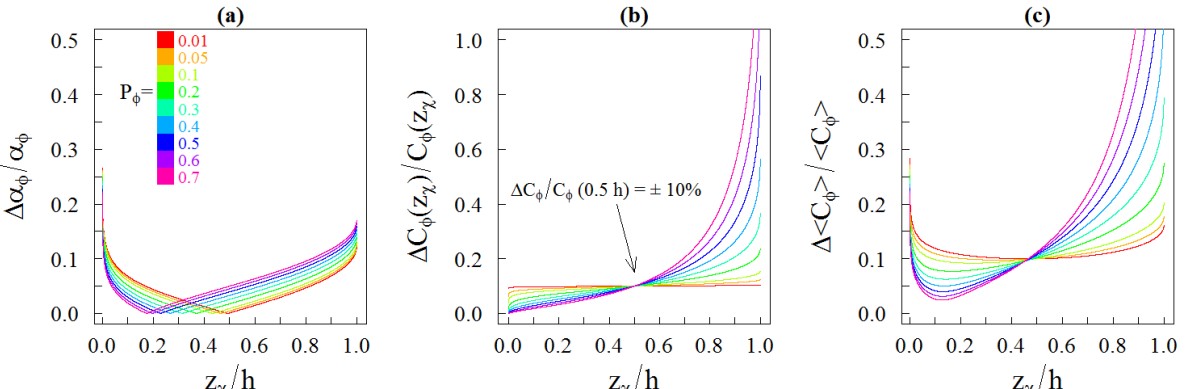

**Figure 11:** (a) Relative error of the predicted $\alpha_\phi$ according to the index sampling height $z_\chi$, for various Rouse numbers $P_\phi$. (b) Relative error of the concentration sampled, inferred from the Zagustin model and assuming $\frac{\Delta C_\phi}{C_\phi}(z_\chi = 0.5\,h) = \pm 10\%$. (c) Relative error on $\langle C_\phi \rangle$ as a function of $z_\chi$.

## 5 Appendices

### A1: Notations

$\langle \ \ \rangle$ = Depth-integrated value

### Main subscripts

$\cdot_\chi$ = Index height value

$\cdot_0$ = Bottom reference height value

$\cdot_r$ = Reference height value

$\cdot_\phi$ = Particle size group $\phi$:

   $\cdot_f$ = Fine sediment particles group ($< 63$ microns)

   $\cdot_s$ = Sand sediment group ($> 63$ microns)

**Terms**

$C$ = Time-averaged Concentration [mg L$^{-1}$]

$d$ = Arithmetic mean diameter [m]

$d_* = d\left(\frac{g\left(\frac{\rho}{\rho_w}-1\right)}{v^2}\right)^{\frac{1}{3}}$ = dimensionless grain size

$g$ = gravitational force [m s$^{-2}$]

$h$ = Mean water depth [m]




$k_s$ = Nikuradse equivalent roughness height [m]

$P$ = Rouse number

$q_s$ = Time-averaged sediment discharge on a vertical [g s$^{-1}$ m$^{-2}$]

$u$ = Time-averaged velocity [m s$^{-1}$]

5   $u_*$ = Shear velocity [m s$^{-1}$]

$w$ = Suspended sediment particle settling velocity [m s$^{-1}$]

$X$ = Mass fraction

$z$ = Height above the bed [m]

$\alpha$ = Ratio between mean concentration and index concentration

10   $\beta$ = Ratio of sediment to eddy diffusivity

$\varepsilon$ = Sediment diffusivity coefficient [m² s$^{-1}$]

$\varepsilon_m$ = Momentum exchange coefficient [m² s$^{-1}$]

$\kappa$ = Constant of Von Kármán

$\upsilon$ = Kinematic viscosity [m² s$^{-1}$]

15   $\rho_w$ = Water density [kg m$^{-3}$]

$\rho$ = Sediment density [kg m$^{-3}$]

$\theta$ = Shield's dimensionless shear stress parameter

$\theta_{cr}$ = Critical dimensionless shear stress threshold

**A2: Soulsby (1997) settling law (terminal velocity):**

$$w = \frac{\upsilon}{d}\left(\sqrt{10.36^2 + 1.049\, d_*^3} - 10.36\right)$$

**A3: Velocity laws:**

**Inner law ("law of the wall"):**

$$u(z) = \frac{u_*}{\kappa}\ln\left(\frac{30\,z}{k_s}\right)$$

**Zagustin (1968) defect law**

$$u(z) = U_{max} - 2 \times \frac{u_*}{\kappa}\operatorname{arctanh}\left(\frac{h-z}{h}\right)^{\frac{3}{2}}$$