# Peer review of "An index concentration method for suspended load monitoring in large rivers of the Amazonian foreland"

_Earth Surface Dynamics, 2018_

## Referee Comment (RC1) · Anonymous Referee #1 · 31 Jan 2019

*General comments*

This manuscript by Santini et al. reports a novel method to link suspended sediment concentration averaged over a whole river channel depth (which is necessary for accurate sediment flux computation), to those ("index") that could be measured at a single point - most commonly at the surface. This is an especially relevant issue for accurately estimating sediment fluxes in larger rivers - which usually display strong vertical gradients in suspended sediment concentration - especially in remote locations where sediment sampling throughout the whole cross-section is not always possible. The manuscript provides a method for a semi-empirical calibration (i.e. the estimate of the

depth-integrated sediment concentration relies on fits of hydraulic-based laws to observed concentrations, while the relationship between depth-averaged and index concentration is empirical) of such relationships, which is shown to be location-dependent, and potentially variable with time across the hydrological year (based on a network of gauging stations located on the largest Amazon tributaries in Peru).

The manuscript is well-written, properly organised, and clearly of great interest to those interested in sedimentary dynamics in large systems or in estimating river suspended sediment fluxes, be it for basic research or for operational reasons. I just think that some readers might be lost - as I was - at some places in the current manuscript, and that it would benefit from further clarification. Comments in this direction are appended below.

*Specific comments*

- p. 3 - l. 4: "thalweg" or simply "bottom"?

- p. 9 - l. 9: Is it "Hypothesis" or "Assumptions"?

- p. 11 - l. 24: "regression slope" should be defined or specified, or better at least better linked to what is said elsewhere (I think it is "alpha"?).

- p. 11 - l. 25: What is meant by "measured alpha"?

- p. 12 - l. 2: Is there a reference for this statement about the delivery of "coarser elements" by the lowland tributaries?

- p. 12 - l. 5: Isn't BOR right at the Andean outlet, and therefore not influenced by lowland inputs?

- p. 12 - l. 7: This is not what I see in Fig. 3... Aren't the axes swapped? And is the color-coding right? The figure and the text sound contradictory, but I might be wrong.

- p. 12 - l. 14: "increases" with what? Downstream?

- p. 12 - l. 28 to - p. 10 - l. 21: That is already a whole lot of interpretation, not only results.

- p. 12 - l. 27: "mixed-load" should be defined.

- p. 12 - l. 28: "random factor" is not clear? Is it meant that this feature appears sort of stochastically?

- p. 12 - l. 29-30: Is this link clearly established, actually?

- p. 13 - l. 25-26: Which "result" exactly are we talking about here? The P-values themselves or their low variability. or the fact that they were averaged per site?

- p. 14 - l. 22: Which "zone"?

- p. 16 - l. 29: Where are the numerical values in the equation from?

- p. 19 - l. 9: Where does this relationship between "" and "h" come from?

- Figure 2: It took me a while to understand what the blue and purple lines are in the upper-left panel - it would be better to clearly indicate what they are within the panel.

- What the x-axis represents is a bit mysterious, as I did not see any explanation in the text. I guess this is because this is a non-dimensional number that is expected to be the same at all locations and in all hydrological conditions?

- Figure 7, caption: What is meant by "observed" and "measured" values here?

- Figure 10: What about the other stations than those shown in panel (b), why aren't they represented? And it would make more sense to me to plot simply "predicted" vs. "measured" (i.e. without the 1.1-factor) and then to plot the 1:1, and/or 1.1:1 line - at least because in this case the caption is wrong, strictly speaking.

*Technical corrections*

- p. 12 - l. 21 (also p. 15 - l. 3): "valid" -> "validate".

- p. 14 - l. 17: "is" -> "are".

- p. 17 - l. 5: "in" -> "at".

- p. 20 - l. 5: Something is missing between "can" and "more".

- p. 20 - l. 7: "the ratio of" -> "the ratio between"?

- p. 20 - l. 11: "ad" -> "and".

- Figure 2, caption: "staked" -> "stacked".

---

## Referee Comment (RC2) · Anonymous Referee #2 · 19 Feb 2019

This manuscript address a key question in our ability to monitor and estimate sediment fluxes in large rivers. Namely, can we use scarse point samples to get appropriate and realistic suspended sediment distribution and flux data? The authors refine exisitng models of suspended sediment profiles to better predict observed sediment concentrations in profiles from a large database from the Amazon basin. The manuscript is well written, justified and prepared, although some additional focus needs to be paid to the figures (see below for details). I think this manuscript will be of great interest for scientisits working in large river environments, and those investing sediment dynamics more broadly.

I would like the authors to clarify a couple of assumptions that underpin their work. On p.8 the authors state the assumption used in this work is that sediment diffusivitiy is concentration-independent. I think it would be worth providing more detial as to the validity of this assumption as it underpins the entirety of the following work. Are there locations (particulalry in large river systems where sediment concentrations are very high - including the Amazon) where diffusivitiy may become dependent on sediment concnetration? What are the implications of this assumption failing on the rest of the work presented by the authors?

Additionally, on p9. the authors make the assumption that the "velocity distribution is vertically uniform" (line 17). This may be the case in straight reaches of large rivers at low and mean flows, but does this assumption hold at the highest of flows where turbulence is likely to be inceased, adn also when highest sediment yields are likely to be evident? Does it also hold for meanders or regions with large bedforms inducing turbulence into the vertical flow profiles? Again, what are the implications of this assumption for the methodology applied and in the results that the authors achieve?

The authors discuss bed-material, but do not detail if bed material samples were directly sampled, or is this assumed to be bed-material in suspension near the bed? If the later, the fact it is in suspension surely classifies it as suspended load? If it is bed-material being transported by traction or other bedload transport mechanisms, this needs clarifying in the text, and raisies some questions as to the applicability of the model to predict truly suspended load, if it is also capturing part of the transported bedload. This may need some explaination and refinement form the authors to make it clear what they mean by this.

Throughout the manuscript the authors refer the inner region of flow. This has quite a few meanings in fluid dynamics and fluvial geomorphology. Could the authors define at the first use what they mean by this term to avoid confusion with the readers.

P.11, line 9: could the authors explain why the velocity profiles are averaged over

60m? How sensitive are the results to this averaging? How does this averaging affect the sediment distributions near river banks that may be <60 m from the bank?

P.12 line 12: The use of the term "random factor" implies some stochasticitiy is incorporated into the model. I'm not sure this is the meaning the authors meant to get across, so i recommend rephrasing to avoid any confusion.

P.14 line 8: Could the authors provide values as to the "small relvative error" in the text? Is this an RMSE, or other error estimate?

P.14 line 17: How likely is it that the velocity at the river bed is zero? Is there completley static flow at the river bed, and if your model predicts this, is it a "reasonable approximation"? Surely there must be flow at the river bed?

P.15 line 5: it would be good to have a discussion about the applicability of this model to other systems. The authors say the model is sensitive to Rouse numbers (especially if large). Are there systems where this may be the case and therefore the model is not applicable? Or is this model only really useful on the Amazon Basin?

Figure 1: Could the authors add error bars to show the range of standard of concentrations measured?

Figure 3: I have trouble differentiating the red and blue scale bar here. I would recommend changing to something less of an issue or those of us who are colour blind. I also note that on p12 line 8, figure 3 is described as showing the "sand mass fraction...on the ratio alpha" whereas figure 3 depicts concentration rather than ratio alpha.

---

## Referee Comment (RC3) · Kathryn Clark (Referee) · 2 Mar 2019

**Reviewer: Kathryn Clark**

**ESurf Discussions**

**An index concentration method for suspended load monitoring in large rivers of the Amazonian foreland**

**Santini et al., 2019**

**Accept with minor revisions**

I really enjoyed reading this paper and learning about the intricacies of proper river SS concentration sampling in large Amazonian rivers. It is evident that the authors have utilised a vast amount of empirical data coupled with a very comprehensive theoretical overview and application. It is evident that a lot of time and effort went into this paper and it was a pleasure to be able to review it.

General Comments

I imagine this type of paper is aimed at specific scientists who know a lot about the topic already. You might be able to reach a wider audience if you remind readers what the terms mean in the sentence, so they don't have to keep flipping to the terms section. For example, page 11 line 7: possibly repeating what ADCP stands for. Another place a term reminder would help could be in the captions on figures. For example, in the caption for figure 11 it might be good to remind readers what h stands for.

Also, there are a lot of stand alone sentences rather than paragraphs throughout the manuscript. Possibly some formatting thought-out could be useful.

The tables and figures were very well done.

During my PhD I collected water samples at the side of the river in the Madre de Dios river, near the Los Amigos tributary. My team collected a water sample every three hours for two weeks, thus it wasn't possible to go to the middle of the river. How useful are these samples for representing the SS concentration of the river?

Specific comments

**Abstract**

Line 22: for the α definition the index and average are switched compared to the terms list

Line 26: choice of using washload over fine. Maybe include washload in the terms section

Line 31: this model *can* be coupled, suggested change to *could likely*, since this wasn't demonstrated in the paper.

**Introduction**

In the first paragraph maybe make specific reference to increase erosion and thus variability in river suspended load?

Line 1: possibly include: climate variability, *specifically extreme rain events*.

Line 7: At least in rivers there would be references to help make this statement stronger.

Line 22: I found it quite a challenge to keep up with what all the terms mean. I would let readers know that there is a terms list in the supplement.

**Materials and Methods**

Line 30: should the line read: water level vs discharge and *concentration* index vs mean concentration?

Line 23: is the fine particle range 0.45 rather than 0.4um?

**Results**

Lines 22, 23: The grammar of this sentence needs work. If needs a *then* in the sentence.

Line 25-27: Isn't it the reverse, that it's important to accurately calculate the concentration of the sand fraction because it is so variable? If the authors want to keep the sentence about the fines, then maybe say: in order to accurately capture the small variations in $\varepsilon$, accurate sampling is key.

Line 9: Should this be referring to figure 6 rather than 7?

Line 18: Check that this is the proper figures reference for this text.

Line 25: suggested grammatical change: very *small* rather than very *low*.

Line 25/26: suggested grammatical change:  Then considering the *poor* contribution. I would suggest changing it to read *small* contribution.

Line 26: possibly change from low *waters* to low *runoff*.

**Discussion**

Line 6: should authors refer to equation 18 when it's not used in the paper?

Where is figure 8b referenced?

Line 5: where is the Rouse number compared to particle size?

Where is figure 10 referenced?

Line 2: Suggested removal of repeated "the concentration"

Line 2: Should "result" read as "*results*"

Line 9: Should $q_{ss}$ just read as $q_s$?

Lines 9-12: If this is the only statement made about the usefulness of remote sensing, I would modify the text in the abstract.

**Appendices**

Line 13: should the fines be from 0.45 to 0.63um?

---

## Author Comment (AC1) · 3 Mar 2019

We thank the referee for his positive appraisal of our manuscript and for the work done to review it. The suggestions of the referee will actually improve the paper.

We address here a first reply to the referee comments before to provide a revised manuscript.

- p. 3 - l. 4: "thalweg" or simply "bottom"?

Here, "bottom" would be more appropriate than "thalweg".

- p. 9 - l. 9: Is it "Hypothesis" or "Assumptions"?

Some of the "assumptions" made in this subsection are further validated by the results. Thus, they can be considered as "hypothesis". However, it is true that a part of the assumption cannot be validated with our data: for example the negligible interaction between the sediment classes. Hence, we are agree to change "hypothesis" for "assumption" in this subsection title, and we thanks the referee for this subtlety of the English language.

- p. 11 - l. 24: "regression slope" should be defined or specified, or better at least better linked to what is said elsewhere (I think it is "alpha"?), and p. 11 - l. 25: What is meant by "measured alpha"?

Fig.2 shows the relations obtained from the field measurements between an index concentration sampled close to the river surface and the cross-sectional average concentration measured with a point sampling method. Thus, here the "measured $\alpha$" refers to the ratio between index and average concentration (i.e. the total sediment concentration), calculated for each measurement, by contrast with the "predicted $\alpha$" calculated with the model from hydraulic parameters (Eq. 15 and 17). For each station, a linear calibration line with a zero intercept was fitted on the measured $\alpha$ values, with a zero intercept (Fig.2). The "regression slope" value fitted on the measured $\alpha$ corresponds to a mean $\alpha$ value for each station and allows comparing the different trends observed on the Fig.2.

In the final reply, this paragraph will be reworded in order to clarify the text.

- p. 12 - l. 28 to - p. 10 - l. 21: That is already a whole lot of interpretation, not only results.

We guess the referee means p. 11, l. 28 to p. 12, l. 21. Yes, this part of the text contains already a lot of interpretations. We believe that the main interest of this part of the paper is to show with observed data that the ratio $\alpha$ between index and average concentration depends on both the basin characteristics and the flow conditions. We can either shorten it, or put it into discussion

- p. 12 - l. 2: Is there a reference for this statement about the delivery of "coarser elements" by the lowland tributaries?

To our knowledge, no reference strictly support the claim that the lowland provides sediments with a coarser grain size than does the central Andean source. This is an assumption partially supported by our grain size data and the following reasoning about the transport capacity (or river competence, i.e. the grain size the river can transport) of the Marañon River, and the main sources of the sediment transported by this river:

The Upper Marañon River mostly drains the Marañon fold and thrust belt in the hollow of the Central Andes, (Pfiffner & Gonzales, 2013), with a lithology mainly sedimentary and roughly exposed to the same rainfall regime.

In the lowland, the left bank tributaries, fueled by an equatorial rainfall regime upon the northern part of the basin, supply almost 55% of the Marañon water discharge. These tributaries drain small Andean areas (which are not part of the central Andes), where igneous rocks predominate and with relatively

low erosion rates (Laraque et al., 2009; Armijos et al., 2013) compared to those observed in the central Andes (i.e. for the Ucayali basin, or the Madre de Dios, Beni and Mamore rivers in Bolivia) (Guyot et al, 2007; Armijos et al., 2013; Pepin et al., 2013; Santini et al., 2014; Latrubresse & Restrepo, 2014; Vauchel et al.; 2017, Calvès et al., 2018).

Thus, we deduced that a substantial part of these tributaries fluxes comes from the wide Cenozoic sedimentary basin located between the Marañon River and the Ecuadorian Andes. This large area, partly drained by the Pastaza and Napo Rivers, experiences a neo-tectonics activity (e.g. the uplift of the Mera Megafan) (Baby et al., 2013, Calvès et al., 2018).

If, until now, none long-term survey has been able to quantify these tributaries inputs, the Napo River sediment budget (Laraque et al., 2009; Armijos et al., 2013b) shows that this lowland area could be the main sediment source for the Marañon's Ecuadorian tributaries crossing it.

This secondary source **could** provide sediments with a coarser grain size than the Central Andean source, as the Napo River does (Table 2). In addition, the transport capacity (slope and water level) of the lower Marañon is higher than the Ucayali's one, thus the river can route this material more easily to its outlet.

Moreover, in the lowland, the floodplain incision mechanically increases the sand mass fraction $X_s$ in the suspended load. Implicitly, the PSD mean diameter shifts with $X_s$, but it does not mean that there is any change in the physical properties (e.g. diameter, density and shape) of the sand fraction. This shift directly affects the ratio $\alpha$, as the vertical concentration gradient depends on the balance between the turbulence strength and the settling velocity (Eq. (4)).

In contrast, the Ucayali and Huallaga's fluxes routed in the lowland mainly originates from the Central Andes, with a predominant contribution of the Eastern Cordillera slopes and its adjacent Sub Andean Zone (e.g. Aalto et al., 2006; Baby et al., 2009; Kober et al., 2015), where the outcrops are mainly Paleozoic and Mesozoic sedimentary rocks (Pfiffner & Gonzales, 2013). Owing to a monsoon-controlled regime, the rainfall spatial distribution may be roughly considered as pseudo-proportional on the basin. Hence, the composition of the mineralogical convoy of this primary source of sediments may be regarded as relatively stable over time, each lithological domain of the Central Andes contributing in a pseudo-proportional manner. Some discrepancies from this general pattern are however due to the variability in land use, hillslope, regolith thickness, soil erodibility, and others secondary factor.

In the floodplain, unlike the Marañon basin, few lateral tributaries come to swell the rivers discharge (Guyot et al, 2007; Armijos et al., 2013; Santini et al., 2014). The nature of the sediment routed through the riverine system depends only of the river transport capacity (weaker than in the Marañon and seasonal) and of the flows back from the floodplain after storage.

**- p. 12 - l. 5: Isn't BOR right at the Andean outlet, and therefore not influenced by lowland inputs?**

The Upper Marañon River and the Santiago River form the Marañon River at BOR. The Santiago River originates in the Ecuadorian Andes, and crosses a piggyback basin before to reach the Marañon River. This area is considered in the text (roughly) as a lowland area. Conversely, the upper Marañon River originates from the central Andes. Recent concentration measurements made in the upper Marañon River (just before the Andean outlet) have shown that $\alpha = \langle C \rangle / C(h) \cong 1.5$ for this river. This value is consistent with the ratios $\alpha = \langle C \rangle / C(h)$ observed in the other rivers having the Central Andes as a main source of sediments, i.e. the Ucayali and Huallaga rivers, and differs from the ratios $\alpha$ observed in the Napo River and the Marañon River at REG, where the lowland contribution is assumed to explain the high ratios $\alpha$ observed (please report to the argumentation exposed in the previous point).

- p. 12 - l. 7: This is not what I see in Fig. 3… Aren't the axes swapped? And is the color-coding right? The figure and the text sound contradictory, but I might be wrong.

According to referee #1 and #2 reports, Fig.3 is quite difficult to read. We propose the following figure instead, where we can see the increase of $\alpha = \langle C \rangle / C(h)$ with the sand mass fraction $X_s$. The concentration uncertainty and the shear stress variability during the hydrological cycle explain the dispersion observed in this figure.

[Figure]

New figure 3: Measured $\alpha = \langle C \rangle / C(h)$ vs sand mass fraction $X_s$. The uncertainty on the concentrations measured and the shear stress variability during the hydrological cycle explain the dispersion observed on the figure.

- p. 12 - l. 14: "increases" with what? Downstream?

Here, we mean that more the particle size is large, more the transport regime may be considered to be capacity-limited, depending only on the available energy to route the sediments.

- p. 12 - l. 27: "mixed-load" should be defined.

 "mixed-load" will be defined in the revised manuscript: The mixed-load is a transition mode of transport, where both bedload and suspended load contribute to the total load in similar proportion. The mixed load regime is found when $\sim 1 < P_\phi < 6$.

- p. 12 - l. 28: "random factor" is not clear? Is it meant that this feature appears sort of stochastically?

Here, we propose to reword "random factor" by "uncertainty factor". Indeed, the term "random factor" was poorly chosen to explain why the coarse bed sediments corresponding to the yellow distribution in Fig.4 complicate the prediction and measurement of the concentration profiles.

Bedforms may affect the process of suspension for the coarser fractions of sands because eddies formed at their leeward side, can swiftly uplift large amounts of this sediments in the water column (fluid burst). Along with the water height and the turbulence structures expansion, the suspension of these coarse sediments becomes rather stochastic than uniform, as their grain size make them very sensitive to the velocity fluctuations, which are stochastic. Thus, their presence is non-permanent and

heterogeneous across the section (and often associated to the movable bed phenomenon affecting the ADCP measurements when none GPS is coupled with this instrument).

Hence, the concentration of this sand fraction is very difficult to measure: at least, such measurement would require a long integration time during the sampling operation to obtain a representative estimate.

Moreover, the suspension of this coarse material may be fundamentally different to the suspension described with a classical diffusion approach (i.e. when considering uniform conditions). Indeed, the stochastic and ephemeral inputs of coarse bed material in the water column is not addressed by the classical suspension theory (inferred from the Prandtl mixing length approximation), which considers time-averaged concentrations and neglects the fluctuation terms (see also the section 4.2, P.17, l.25 to 27).

However, such process is rather limited to the first heights of the water column and concern a small fraction of the total load in suspension (when a long enough timescale is considered, which is usually the case for basic sediment load assessments in large rivers). This is why it is rather an "uncertainty factor" than a "random factor".

- p. 12 - l. 29-30: Is this link clearly established, actually?

This follows up the previous point. During the fitting procedure of suspension models on the observed concentration profiles, we noticed that some profiles were more disturbed (i.e. more dispersed) on the lower half of the water column for high flow conditions than for normal flow conditions. We related these concentration fluctuations to a more important influence of the coarse bed sands uplifted in suspension (yellow distribution in Fig.4) on the measured concentration.

The following figure show how the coarse sand fraction can affect the Rouse number value fitted on concentration profiles. The black circles are observed sand concentrations over depth on a sampled vertical at TAM. At this vertical, a movable bed phenomenon was detected during the ADCP measurement. The continue line in black correspond to a Rousean profile fitted on the concentrations measured. A second Rousean profile was fitted on the three first points of the water depth (pink line, which would correspond to the pink distribution in Fig. 4a). The yellow profile results from the difference between the two previous ones. Assuming that no coarse sediments are present in the upper part of the flow, the yellow profile would correspond to the mixed-load represented by the yellow distribution in Fig. 4a.

In this example, the mixed-load weight moderately on the Rouse number fitted on the sand concentration profile (here, the Rouse number for the pink profile is equal to 0.33, and equal to 0.4 for the black profile). However, when considering a mean concentration profile for the whole cross-section, the mixed-load influence decrease, as its presence is heterogeneous across the section.

[Figure]

- p. 13 - l. 25-26: Which "result" exactly are we talking about here? The P-values themselves or their low variability. Or the fact that they were averaged per site?

We guess the referee means p. 13, l. 15-16. The low variability of the Rouse numbers fitted on the concentration profiles indicates there is a kind of dynamic equilibrium between the sediment settling velocity and the shear stress under nominal flow conditions, although some extreme Rouse number values were measured during severe drought events at the lowland stations.

This is the flow zone near the water surface ($\alpha_s(h, P_s)$). We will reword the sentence in the revised manuscript in order to clarify the text.

- p. 16 - l. 29: Where are the numerical values in the equation from?

The original Rose and Thorne (2001) law $\beta_\phi = 3.1 \exp\left(-0.17 \, \frac{u_*}{w_\phi}\right)$ is an empirical function based upon a regression analysis on measured concentration profiles. The factor 3.1 of Eq. 19 is coming from this law.

In the present work, the following modification of the Rose and Thorne (2001) law is proposed:

$$\beta_\phi = 3.1 \exp\left(-0.19 \, \frac{u_*\left(\frac{h}{d_s}\right)^{0.6}}{1000 \times w_\phi}\right) + 0.16$$

where the numerical values of Eq. 19 were fitted to obtain the best agreement between the $\beta_\phi$ inferred from the measured concentration profiles (Table 2) and the $\beta_\phi$ predicted by Eq.19.

The following correction of the $\left(\frac{u_*}{w_\phi}\right)$ ratio allows reducing the discrepancy observed in (Fig. 8a) between the Rose and Thorne law and the observed $\beta_\phi$ values:

$$\left(\frac{u_*}{w_\phi}\right)_{corrected} = \frac{0.19}{0.17} \frac{\left(\frac{h}{d_s}\right)^{0.6}}{1000} \left(\frac{u_*}{w_\phi}\right)$$

The original Rose and Thorne law tends to zero for low values of the ratio $\left(\frac{u_*}{w_\phi}\right)$ (Fig. 8a). Adding a value of 0.16 allows to take into account the $\beta_f$ for the fine sediments in the modified law (Fig. 8b).

When the stream's velocity profile is logarithmic in nature $\left(u(z) = \frac{u_*}{\kappa}\ln\left(\frac{z}{z_0}\right)\right)$, it can be shown that the depth-integrated velocity $\langle u \rangle$ is equal to: $\langle u \rangle \cong \frac{h}{e} \cong 0.37h$:

$$\int_{z_0}^{h} \frac{u_*}{\kappa}\ln\left(\frac{z}{z_0}\right) dz \cong \frac{u_*}{\kappa}(\ln(h) - 1) - \frac{u_*}{\kappa}\ln(z_0) \cong \frac{u_*}{\kappa}\ln\left(\frac{\frac{h}{e}}{z_0}\right) = u\left(\frac{h}{e}\right), \text{ with } \ln(e) = 1 \text{ and } h \gg z_0$$

It is a result well known by the hydrologist measuring river discharges with a classic current-meter. Other points of interest exist on the velocity profile: Prony first shown that usually $\langle u \rangle \cong u(0.8h - 0.9h)$. The USGS recommends to deploy a two points method: $\langle u \rangle \cong \frac{1}{2}(u(0.8h) + u(0.2h)))$. This method could be also coupled with Eq. 21 in order to define a protocol for concentration measurement.

When the velocity profile is not logarithmic (which is often the case in small streams with high Froude numbers), a three-point method can also be deployed (0.8h, 0.4h, 0.2h).

- Figure 2: It took me a while to understand what the blue and purple lines are in the upper-left panel
- it would be better to clearly indicate what they are within the panel.

[Figure]
* * *
Please find below a new figure 2 with a short text in the upper left panel:

[Figure]

- What the x-axis represents is a bit mysterious, as I did not see any explanation in the text. I guess this is because this is a non-dimensional number that is expected to be the same at all locations and in all hydrological conditions?

We guess it is the x-axis of Fig. 5 and not Fig. 2? If it is Fig. 5, yes, it is a non-dimensional number, and the diffusivity profile is expected to follow the theoretical models (Camenen & Larson, Rouse, Van Rijn or Zagustin) at all locations and in all hydrological conditions.

- Figure 7, caption: What is meant by "observed" and "measured" values here?

For each site, mean $\alpha_f$ and $\alpha_s$ were computed according to Eq. 17 with the mean parameters given in table 2. Then, "predicted $\alpha$" were calculated from Eq. 15 ($\alpha = X_f\,\alpha_f + X_s\,\alpha_s$), by considering the mean $X_f$ and $X_s$ values also given in table 2 and calculated from the concentration measurements. Finally, the "predicted $\alpha$" are compared in Fig.7 with the "observed $\alpha$", i.e. the mean ratios between index and average total concentration, measured at each site. The "observed $\alpha$" corresponds to the slope of the regression lines in Fig. 2.

Maybe we could change "observed $\alpha$" for "measured $\alpha$" both in the ms and in the figure 7?

- Figure 10: What about the other stations than those shown in panel (b), why aren't they represented? And it would make more sense to me to plot simply "predicted" vs. "measured" (i.e. without the 1.1-factor) and then to plot the 1:1, and/or 1.1:1 line – at least because in this case the caption is wrong, strictly speaking.

Samples for the characterization of the bed material PSD were collected only in four sites: BEL (2 PSD), REQ (1 PSD), REG (1 PSD) and TAM (2 PSD). Please note that for TAM and BEL, only the average diameter of the two PSD measured is plotted (However, we could plot all the measurements if necessary). There is no PSD measurements for the bed material at BOR, LAG, PIN and CHA. We will add

some words in the revised ms about the collection of bed material in the "material and methods" section.

Please find below the new figure 10 with the modification asked:

[Figure]

*Technical corrections*

- p. 12 - l. 21 (also p. 15 - l. 3): "valid" -> "validate".

- p. 14 - l. 17: "is" -> "are".

- p. 17 - l. 5: "in" -> "at".

- p. 20 - l. 5: Something is missing between "can" and "more".

- p. 20 - l. 7: "the ratio of" -> "the ratio between"?

- p. 20 - l. 11: "ad" -> "and".

- Figure 2, caption: "staked" -> "stacked".

Many thanks for the technical corrections; we will incorporate them in the revised ms.

Sincerely,

William Santini

**References:**

Aalto, R., Dunne, T., Guyot, J. L.: Geomorphic Controls on Andean Denudation Rates, The Journal of Geology 114(1), 85-99, doi:10.1086/498101, 2006.

Armijos, E., Laraque, A., Barba, S., Bourrel, L., Ceron, C., Lagane, C., Magat, P., Moquet, J. S., Pombosa, R., Sondag, F., Vauchel, P., Vera, A., Guyot, J. L.: Yields of suspended sediment and dissolved solids from the Andean basins of Ecuador, Hydrological Sciences Journal, 58(7), 1478-1494, doi:10.1080/02626667.2013.826359, 2013b.

Baby, P., Guyot, J. L.:Tectonic control of erosion and sedimentation in the Amazon Basin of Bolivia, Hydrological Processes, 23(22), 3225-3229, doi:10.1002/hyp, 2009.

Baby, P., Rivadeneira, M., Barragan, R., Christophoul, F.: Thick-skinned tectonics in the Oriente foreland basin of Ecuador, Geological Society, London, Special Publications 377(1), 59-76, doi:10.1144/SP377.1, 2013.

Calvès, G., Calderon, Y., Hurtado Enriquez, C., Brusset, S., Santini, W., Baby, P.: Mass Balance of Cenozoic Andes-Amazon Source to Sink System, Marañon Basin, Peru, Geosciences 8(5), 167, doi:10.3390/geosciences8050167, 2018.

Kober, F., Zeilinger, G., Hippe, K., Marc, O., Lendzioch, T., Grischott, R., Christl, M., Kubik, P., Zola, R.: Tectonophysics Tectonic and lithological controls on denudation rates in the central Bolivian Andes, Tectonophysics 657, 230-244, doi:10.1016/j.tecto.2015.06.037, 2015.

Latrubesse, E. M., Restrepo, J. D.: Sediment yield along the Andes: continental budget, regional variations, and comparisons with other basins from orogenic mountain belts, Geomorphology 216, 225-233, doi:10.1016/j.geomorph.2014.04.007, 2014.

Pepin, E., Guyot, J. L., Armijos, E., Bazan, H., Fraizy, P., Moquet, J. S., Noriega, L., Lavado, W., Pombosa, R., Vauchel, P.: Climatic control on eastern Andean denudation rates (Central Cordillera from Ecuador to Bolivia), Journal of South American Earth Sciences, 44, 85-93, doi:10.1016/j.jsames.2012.12.010, 2013.

Pfiffner, A. O., Gonzalez, L.: Mesozoic–Cenozoic Evolution of the Western Margin of South America: Case Study of the Peruvian Andes, Geosciences 3(2), 262–310, doi:10.3390/geosciences3020262, 2013.

---

## Editor Comment (EC1) · Robert Hilton (Editor) · 29 Mar 2019

Dear authors,

Your manuscript has now been seen by three referees. They all highlight the general quality, interest, importance, and good fit with ESurf themes. However, they call for important clarifications and in my opinion the manuscript requires moderate/substantial revisions that address those before consideration for final publication.

Please respond in full to their comments in the discussion forum, and then provide a revised manuscript for review.

[Figure]

Best regards, Bob Hilton AE ESurf

**ESurfD**

---

## Author Comment (AC2) · 30 Mar 2019

**Response to review of referee #1**

William Santini, Benoît Camenen, Jérôme Le Coz, Philippe Vauchel and Jean-Michel Martinez

15/03/2019

We thank the referee#1 for her/his positive appraisal of our manuscript and thoughtful review: the suggestions of the referee will actually improve the manuscript.

We attached a marked version of the manuscript, highlighting all modifications in yellow.

Below we list the referee comments in italic font and our responses in regular font.

Sincerely,

W. Santini et al.

**General Comments**

*This manuscript by Santini et al. reports a novel method to link suspended sediment concentration averaged over a whole river channel depth (which is necessary for accurate sediment flux computation), to those ("index") that could be measured at a single point - most commonly at the surface. This is an especially relevant issue for accurately estimating sediment fluxes in larger rivers - which usually display strong vertical gradients in suspended sediment concentration - especially in remote locations where sediment sampling throughout the whole cross-section is not always possible. The manuscript provides a method for a semi-empirical calibration (i.e. the estimate of thedepth-integrated sediment concentration relies on fits of hydraulic-based laws to observed concentrations, while the relationship between depth-averaged and index concentration is empirical) of such relationships, which is shown to be location-dependent, and potentially variable with time across the hydrological year (based on a network of gauging stations located on the largest Amazon tributaries in Peru).*

*The manuscript is well-written, properly organised, and clearly of great interest to those interested in sedimentary dynamics in large systems or in estimating river suspended sediment fluxes, be it for basic research or for operational reasons. I just think that some readers might be lost - as I was - at some places in the current manuscript, and that it would benefit from further clarification. Comments in this direction are appended below.*

**Specific comments**
* * *
**Introduction**
* * *
*- p. 3 - l. 4: "thalweg" or simply "bottom"?*

Thank you for this suggestion. Yes, here "bottom" would be more appropriate than "thalweg". The sentence was modified in the marked manuscript (p.3, l.8).
* * *
**Materials and Methods**
* * *
*- p. 9 - l. 9: Is it "Hypothesis" or "Assumptions"?*

Some of the "assumptions" made in this subsection are further validated by the results. Thus, they can be considered as "hypothesis". On the other hand, however, a part of the assumption cannot be validated with our data: for example, the negligible interaction between the sediment classes.

Hence, we changed "hypothesis" for "assumptions" in this subsection title (p.9, l.13).
* * *
**Results**
* * *
*- p. 11 - l. 24: "regression slope" should be defined or specified, or better at least better linked to what is said elsewhere (I think it is "alpha"?), and p. 11 - l. 25: What is meant by "measured alpha"?*

Fig.2 shows the regressions obtained from the field measurements between an index concentration sampled close to the river surface and the cross-sectional average concentration measured with a point sampling method. Thus, the "measured $\alpha$" refers to the ratio between index and average concentration (i.e. the total sediment concentration), calculated for each measurement, by contrast with the "predicted $\alpha$" calculated with the model from hydraulic parameters (Eq. 15 and 17).

The regression slope corresponds indeed to a fit to the observed $\alpha$-values. We skip this term to avoid any confusion

In the marked ms, we rewrote the section 3.1.1 (p.12, l.4 to p.12, l.22).

Moreover, we rephrased the following sentences and captions to avoid a potentially inappropriate use of the word "measured":

- **P.4, l.24:** "the ratios $\alpha = \langle C \rangle / C(z_\chi)$ observed at 8 gauging stations"
- **P.12, l.4 and 10:** "The observed $\alpha$"
- **P.16, l.6:** "the $P_\phi$ fitted on measured concentration profiles"
- **P.16, l.23-24:** "the $\beta_\phi$ inferred in this study from measured profiles of concentration, particle diameter and velocity"
- **P.17, l.16:** "the observed $\beta_\phi$"
- **Caption for figure 2:** "Observed ratios $\alpha = \langle C \rangle / C(h)$"
- **Caption for figure 3:** "Observed ratios $\alpha = \langle C \rangle / C(h)$"
- **Caption for figure 7:** "observed $\alpha_s$ values"
* * *
*- p. 12 - l. 28 to - p. 10 - l. 21: That is already a whole lot of interpretation, not only results.*

We guess the referee means p.11, l.28 to p.12, l.21.

Yes, the section 3.1.1 contains some interpretation of the observed data. Indeed, the main interest of this section is to show that the ratio $\alpha$ between index and average concentration depends on both the basin characteristics and the flow conditions. Moreover, the results in this section highlights the key challenge of providing a proper model of the PSD using a limited number of sediment classes, and validate the discrete approach proposed to model $\alpha$.

In order to ensure not to lose the reader, and to explain the usefulness of the hydraulic insight in this study, we would prefer to keep the section 3.1.1 where it is.

Note that we rephrased the section 3.1.1.
* * *
*- p. 12 - l. 2: Is there a reference for this statement about the delivery of "coarser elements" by the lowland tributaries?*

To our knowledge, no reference strictly supports the claim that the lowland provides sediments with a coarser grain size than does the central Andean source. This is an assumption based on our grain size data and on the Napo River sediment budget (Laraque et al., 2009; Armijos et al., 2013), which shows that the lowland area is the main sediment source in this basin.

Section 3.1.1. was rewritten in the revised ms.
* * *
*- p. 12 - l. 5: Isn't BOR right at the Andean outlet, and therefore not influenced by lowland inputs?*

The Upper Marañon River and the Santiago River form the Marañon River at BOR. The Santiago River originates in the Ecuadorian Andes, and crosses a piggyback basin before reaching the Marañon River. This area is a lowland area.

Section 3.1.1. was rewritten in the revised ms to take into account this remark: "The river incision of this secondary source, and/or the Ecuadorian Andes, could provide coarser elements than does the central Andean source and explain why the ratios $\alpha$ are higher at BEL and REG than at the other sites."
* * *
*- p. 12 - l. 7: This is not what I see in Fig. 3... Aren't the axes swapped? And is the color-coding right? The figure and the text sound contradictory, but I might be wrong.*

According to referee #1 and #2 reports, Fig.3 is indeed quite difficult to read. We modified the figure in order to better detect the increase of $\alpha = \langle C \rangle / C(h)$ with sand mass fraction $X_s$. The concentration measurement uncertainty and the shear stress variability during the hydrological cycle explain the observed dispersion.

[Figure]

**New figure 3:** Observed ratios $\alpha = \langle C \rangle / C(h)$ of total mean concentration to the total index concentration sampled at the water surface vs sand mass fraction $X_s$.

*- p. 12 - l. 14: "increases" with what? Downstream?*

We rephrased the sentence in the marked ms (p.12, l.29): "On the other hand, sand transport regime is capacity-limited, depending only on the available energy to route the sediments. Since the flow energy significantly decreases with the decreasing bed slope, sand total load is gradually decoupled from the washload in the floodplain, and the washload concentration is no longer a good proxy of the coarse particle concentration"
* * *
*- p. 12 - l. 27: "mixed-load" should be defined.*

The mixed-load is a transition mode of transport, where both bedload and suspended load contribute to the total load in similar proportion. The mixed load regime is found when $\sim 1 < P_\phi < 6$.
We skipped this term and the section 3.1.2 has been rephrased (p.13, l.10 − 15).
* * *
*- p. 12 - l. 28: "random factor" is not clear? Is it meant that this feature appears sort of stochastically?*

We agree with the reviewer that the sentence is unclear. We skipped this term in the revised ms, and the section 3.1.2 has been rephrased (p.13, l.10 − 15).
* * *
*- p. 12 - l. 29-30: Is this link clearly established, actually?*

Fig. 4 shows that one can easily observe vertical gradients for both the concentration and grain size. As a consequence, a proper modelling of this gradients would require a multi-class description of the sediment.

This section has been rephrased in the revised ms (p.13, l.12 − 13).
* * *
*- p. 13 - l. 25-26: Which "result" exactly are we talking about here? The P-values themselves or their low variability. Or the fact that they were averaged per site?*

We guess the referee means p.13, l.15-16.
We have rephrased the sentence in the revised ms: "this result" becomes "this low variability" (p.13, l.31).

*- p. 14 - l. 22: Which "zone"?*

This is the flow zone near the water surface. The sentence has been reworded in the marked manuscript in order to clarify the text (p.15, l.9).

**Discussion**

*- p. 16 - l. 29: Where are the numerical values in the equation from?*

We have added in the revised ms (p.17, l.24 – 25) the following sentence: "[...] where the coefficient 3.1 comes from the Rose and Thorne law (2001). Other numerical values in Eq. (19) were fitted to obtain the best agreement with the $\beta\_\phi$ inferred from the measured concentration profiles (Table 2)".
Moreover, we have rewritten (p.17, l.23) more clearly the equation 19 as:

$$\beta_\phi = 3.1 \exp\left[-0.19 \times 10^{-3} \frac{u_*}{w_\phi} \left(\frac{h}{d_s}\right)^{0.6}\right] + 0.16\,, \tag{19}$$

*- p. 19 - l. 9: Where does this relationship between "" and "h" come from?*

Assuming the stream's velocity profile can be modeled by the logarithmic profile $\left(u(z) = \frac{u_*}{\kappa}\ln\left(\frac{z}{z_0}\right)\right)$, it can be shown that the depth-integrated velocity $\langle u \rangle$ is approximatively equal to: $\langle u \rangle \cong \frac{h}{e} \cong 0.37h$:

$$\int_{z_0}^{h} \frac{u_*}{\kappa}\ln\left(\frac{z}{z_0}\right) dz \cong \frac{u_*}{\kappa}(\ln(h) - 1) - \frac{u_*}{\kappa}\ln(z_0) \cong \frac{u_*}{\kappa}\ln\left(\frac{\frac{h}{e}}{z_0}\right) = u\left(\frac{h}{e}\right), \text{ with } \ln(e) = 1 \text{ and } h \gg z_0$$

The sentence was rephrased in the revised ms (p.20, l.9).

**Figures**

*- Figure 2: It took me a while to understand what the blue and purple lines are in the upper-left panel - it would be better to clearly indicate what they are within the panel.*

Please find below a new figure 2 with a short text in the upper left panel. This figure has been added to the revised ms.

[Figure]

**New figure 2:** Observed ratios $\alpha = \langle C \rangle / C(h)$ of total mean concentration to the total surface concentration, stacked by river basin, with trend lines. For the Amazonas River basin at TAM (upper-left panel), the REG and REQ trend lines were reported. Dashed lines: first bisector.
* * *
*- What the x-axis represents is a bit mysterious, as I did not see any explanation in the text. I guess this is because this is a non-dimensional number that is expected to be the same at all locations and in all hydrological conditions?*

**We guess it is the x-axis of Fig. 5 and not Fig. 2?** If it is Fig. 5, yes, it is a non-dimensional number, and yes, the diffusivity profile is expected to follow the theoretical models (Camenen & Larson, Rouse, Van Rijn or Zagustin) at all locations and in all hydrological conditions.
The caption for the figure 5 was modified: "Dimensionless sediment diffusivity coefficient derived from the measured concentration profiles"
* * *
*- Figure 7, caption: What is meant by "observed" and "measured" values here?*

For the ratios $\alpha$, all the terms "measured" were replaced by "observed" in the ms to avoid any confusion.

For each site, mean $\alpha_f$ and $\alpha_s$ were computed according to Eq. 17 with the mean parameters given in table 2. Then, "predicted $\alpha$" were calculated from Eq. 15 ($\alpha = X_f\,\alpha_f + X_s\,\alpha_s$), by considering the mean $X_f$ and $X_s$ values also given in table 2 and calculated from the concentration measurements. Finally, the "predicted $\alpha$" are compared in Fig.7 with the "observed $\alpha$", i.e. the mean ratios between index and average total concentration, observed at each site.
* * *
*- Figure 10: What about the other stations than those shown in panel (b), why aren't they represented? And it would make more sense to me to plot simply "predicted" vs. "measured" (i.e. without the 1.1-factor) and then to plot the 1:1, and/or 1.1:1 line – at least because in this case the caption is wrong, strictly speaking.*

Samples for the characterization of the bed material PSD were collected only at four sites: BEL (2 PSD), REQ (1 PSD), REG (1 PSD) and TAM (2 PSD). Please note that for TAM and BEL, only the average diameter of the two PSD measured is plotted. There are no PSD measurements for the bed material at BOR, LAG, PIN and CHA.

P.6, l.16-17, we improved the description in the revised ms about the collection of bed material in the "material and methods" section (p.6, l.13 - 14).

Please find below the new figure 10 with the modification asked; this figure has been added to the revised ms.

[Figure]

**Figure 10:** (a) Prediction of the mean diameter $d_s$ at the TAM gauging station for $u_* = 0.12\ \mathrm{m\ s^{-1}}$ (mean flow conditions) (b) Predicted vs measured $d_s$ at BEL, REQ, REG and TAM. Dashed line: first bisector. Continue line: best fit.

**Technical corrections**

*- p. 12 - l. 21 (also p. 15 - l. 3): "valid" -> "validate".*

*- p. 14 - l. 17: "is" -> "are".*

*- p. 17 - l. 5: "in" -> "at".*

*- p. 20 - l. 5: Something is missing between "can" and "more".*

*- p. 20 - l. 7: "the ratio of" -> "the ratio between"?*

*- p. 20 - l. 11: "ad" -> "and".*

*- Figure 2, caption: "staked" -> "stacked".*

Many thanks for these corrections; we have revised all the corrections in the ms.

**Additional references:**

Armijos, E., Laraque, A., Barba, S., Bourrel, L., Ceron, C., Lagane, C., Magat, P., Moquet, J. S., Pombosa, R., Sondag, F., Vauchel, P., Vera, A., Guyot, J. L.: Yields of suspended sediment and dissolved solids from the Andean basins of Ecuador, Hydrological Sciences Journal, 58(7), 1478-1494, doi:10.1080/02626667.2013.826359, 2013b.

[revised manuscript text omitted]

---

## Author Comment (AC3) · 30 Mar 2019

**Response to review of referee #2**

William Santini, Benoît Camenen, Jérôme Le Coz, Philippe Vauchel and Jean-Michel Martinez

15/03/2019

We thank referee #2 for her/his positive appraisal of our manuscript and the thorough review done. The suggestions of the referee will improve the paper.

We attached a marked version of the manuscript, highlighting all modifications in green. Below we list the referee comments in italic font and our responses in regular font.

Sincerely,

W. Santini et al.

**General Comments**

*This manuscript address a key question in our ability to monitor and estimate sediment fluxes in large rivers. Namely, can we use scarse point samples to get appropriate and realistic suspended sediment distribution and flux data? The authors refine existing models of suspended sediment profiles to better predict observed sediment concentrations in profiles from a large database from the Amazon basin. The manuscript is well written, justified and prepared, although some additional focus needs to be paid to the figures (see below for details). I think this manuscript will be of great interest for scientisits working in large river environments, and those investing sediment dynamics more broadly.*

**Specific comments**

*I would like the authors to clarify a couple of assumptions that underpin their work. On p.8 the authors state the assumption used in this work is that sediment diffusivitiy is concentration-independent. I think it would be worth providing more detial as to the validity of this assumption as it underpins the entirity of the following work. Are there locations (particulalry in large river systems where sediment concentrations are very high - including the Amazon) where diffusivitiy may become dependent on sediment concnetration? What are the implications of this assumption failing on the rest of the work presented by the authors?*

Thank you for raising this very interesting issue about the effect of particle concentration on sediment diffusivity, which is a long-standing problem for the scientific community interested in the sediment transport (e.g. Van Rijn, 1984; Graf and Cellino, 2002; Pal and Ghoshal, 2016; Gualtieri et al. 2017).

P.8, l.25, we effectively assume that the sediment diffusivity $\varepsilon_\phi$ is concentration-independent as many authors did (Van Rijn, 194, Camenen & Larson, 2008, among others). The coefficient $\varphi$ introduced by Van Rijn (1984) to account for the dampening of the fluid turbulence by the sediment particles is indeed neglected in most models. Its influence on the sediment diffusivity would be visible for high concentrations only (C>10g/L) that never occurred in the lowland Amazonian rivers. Also, any empirical equation for $\beta_\phi$ as the one proposed here could intrinsically include effects of damping as it is fitted to experimental data.

Moreover, the data do not show relationship between the concentration and $\beta_\phi$ (Fig. 8b or Fig. RC2_1).

Anyway, the concentration influence on the sediment diffusivity could be a secondary factor for the large Amazonian Rivers, such as the particle characteristics (shape, grain size, density...), the aggregation phenomenon, or the level of stratification of the flow (Pal and Ghoshal, 2016; Gualtieri et al. 2017). Unfortunately, the uncertainties of our dataset collected in field conditions do not allow us to further investigate the influence of these secondary order factors on the sediment diffusivity.

[Figure]

**Figure RC2_1:** Ratio $\beta_\phi$ of sediment diffusivity to momentum diffusivity vs total concentration.

Changes in the manuscript:

- We add a paragraph in the discussion section, p.17, l.5 - 9 and two more references (Pal and Ghoshal, 2016; Gualtieri et al. 2017).

- Additionally, we have modified the sentence p.9, l.3 – 4: "This coefficient value is equal to the unity if the sediment diffusion $\varepsilon\phi$ distribution is concentration-independent, which was an assumption used in the present work **because of the range of concentration measured in the Amazonian lowland Rivers (Table 1), and discussed further in section 4.1."**
* * *
*Additionally, on p9. the authors make the assumption that the "velocity distribution is vertically uniform" (line 17). This may be the case in straight reaches of large rivers at low and mean flows, but does this assumption hold at the highest of flows where turbulence is likely to be inceased, adn also when highest sediment yields are likely to be evident? Does it also hold for meanders or regions with large bedforms inducing turbulence into the vertical flow profiles? Again, what are the implications of this assumption for the methodology applied and in the results that the authors achieve?*

1) About the assumption: "velocity distribution is vertically uniform" and the implications of this assumption for the methodology applied

In this work, the mean concentration $\langle C \rangle$ [mg L⁻¹] is defined as (Eq. (1)):

$$\langle C \rangle = \frac{\int_{z_0}^{h} C(z) \times u(z) dz}{\int_{z_0}^{h} u(z) dz},$$

where $C(z)$ is a cross-sectional averaged profile. The depth-integration of this concentration profile is weighted by the velocity distribution on the water depth.

By "the velocity distribution is vertically uniform", we first mean that the velocity gradients are negligible in comparison to the sand concentration gradients on the water column (Fig. RC2_2). Indeed, for the range of nominal Rouse numbers considered in this work $(P_\phi < 0.6)$, the bottom concentration gradient has little influence on $\langle C_\phi \rangle$ because the velocity strongly decrease in this region of the flow $(\sim [z_0 - 0.1\,h])$. Moreover, for these regimes of suspension, the main part of the sediment load is transported above this particular region of the flow. Hence, the influence of the velocity distribution on the depth-averaged concentration can be neglected in Eq.(1).

[Figure]

**Figure RC2_2:** (a) comparison between a typical velocity gradient (red line) and concentration gradient for various Rouse numbers. (b) ratio of concentration to velocity gradient with depth.

Conversely, for sediment profiles with higher Rouse numbers $(P_\phi \geq 0.6)$, the main bulk of sediment is transported near the riverbed, where the velocity distribution experiences its strongest gradient. Then, it is necessary to weight the depth-integrated concentration by the velocity distribution. The higher the Rouse number, the more it will be necessary to describe accurately the profiles of concentration and velocity near the riverbed (Fig. RC2_3) to calculate $\langle C_\phi \rangle$.

As for the fine fraction, the concentration profiles are fairly uniform ($P_f < 0.1$). The mean concentration $\langle C_f \rangle$ is then quasi-independent to the vertical velocity distribution (Fig. RC2_3).

[Figure]

**Figure RC2_3:** Discrepancy between a mean concentration weighted by velocity distribution and a mean concentration calculated without velocity weighting. Two cases of velocity distributions were considered here: one for Ks = 0.01 m (triangles) and another for Ks = 0.1 m (crosses).

The sub-section 2.3.1 was rephrased in the manuscript (p.9, l.26 to p.10, l.2 of the marked ms).

2) About the influence of bedforms and meanders on the velocity profiles

Thank you again for raising another interesting point.

Usually, flows in large rivers are gradually varied. It means that the changes in flow conditions are very progressive, even in meanders, which are very large in the Amazonian context.
In the following figure (Fig. RC2_4), we can see that the secondary velocities in a typical cross-section of the Amazonas River with meander are one to two orders of magnitude smaller than the streamwise flow velocities.

[Figure]

**Figure RC2_4:** Contour of streamwise velocity and secondary velocities (arrows) in a typical cross-section of the Amazonas River with meander (output of the software Velocity Mapping Toolbox of the USGS).

In natural rivers, the bedforms are usually eroded during the rising flows, if the flow is able to peel off the particles of the bed. When the stream power becomes weaker, an aggradation with finer sandy fractions take place, especially close to the banks, but usually without modifying seriously the major bed configuration. This seasonal breathing of the riverbed can leads to changes in flow resistance values (i.e. in form drag), and therefore affect the shape of the velocity profiles.
During the ADCP data processing, the shape of the velocity profiles ($u(z)/u(0.37\,h)$) showed low variability according to the hydrological conditions, even during the highest flows (Fig. RC2_5). This observation supports the assumption that the cross-sectional bed roughness variations (skin + form roughness) during the hydrological cycle were small in the cross-section considered in this study. However, the velocity profiles with higher flow conditions were straighter than with low flow conditions, suggesting moderate changes in bed roughness (Fig. RC2_5).

[Figure]

**Figure RC2_5:** Velocity profiles normalized by the velocity at **0.37 $h$**, at REQ, Ucayali River, for various flow conditions.

Nevertheless, these variations of bed roughness in the hydrological cycle are too small, as well as the potential effect of the meanders on the velocity profiles, to affect seriously the calculation of the mean concentration $\langle C_\phi \rangle$ in the range of $P_\phi$ considered (Fig. RC2_3).

Thus, the changes in the section sub-section 2.3.1 (p.9, l.26 to p.10, l.2) should cover all the intricacies related to the assumption on the velocity profiles (Please note that the gradually varied regime is mentioned in section 2.1, P.5, l.16 of the marked ms).
* * *
*The authors discuss bed-material, but do not detail if bed material samples were directly sampled, or is this assumed to be bed-material in suspension near the bed? If the later, the fact it is in suspension surely classifies it as suspended load? If it is bed-material being transported by traction or other bedload transport mechanisms, this needs clarifying in the text, and raisies some questions as to the applicability of the model to predict truly suspended load, if it is also capturing part of the transported bedload. This may need some explaination and refinement form the authors to make it clear what they mean by this.*

The samples for the characterization of the bed material PSD were collected at four sites: BEL, REQ, REG and TAM. The bed material was dragged on the riverbed.

P.6, l.21 – 22, we improved the description in the revised ms about the collection of bed material in the "material and methods" section.

*Throughout the manuscript the authors refer the inner region of flow. This has quite a few meanings in fluid dynamics and fluvial geomorphology. Could the authors define at the first use what they mean by this term to avoid confusion with the readers.*

Thanks for this suggestion. We have added a definition of the inner region of the flow p.11, l.29: "The imprecise knowledge of the exact bed elevation, the side lobe interferences, the beam angle, which induces a large measurement area, and the instrument's pitch and roll all cause the ADCP velocity data to be inaccurate in the inner flow region (i.e. the region of the flow under bed influence $\sim [z_0, 0.2\, h]$)."
* * *
*P.11, line 9: could the authors explain why the velocity profiles are averaged over 60m? How sensitive are the results to this averaging? How does this averaging affect the sediment distributions near river banks that may be <60 m from the bank?*

1) Why the velocity profiles are averaged over 60m:

The ADCP velocity profiles were averaged over a spanwise length of ~60 m around each concentration profile position in order to dampen the influence of velocity fluctuations, local perturbations, and ADCP measurement errors. Indeed, in this work, we chose to consider flow conditions at a reach scale rather than point-to-point flow conditions.

To obtain representative velocity profiles, and to derive robust $u_*$ values from these profiles, we found that about 30 ADCP "ensembles" (i.e. measurement verticals of velocity) were required, corresponding to a spatial average of about 40 to 70 m in the cross-sectional direction. This was consistent with the methodology applied by Armijos et al. (2016) (50-60 ensembles, corresponding to 10% of the total width of the section) or Lupker et al., 2011 (30 ensembles, 40 to 70 m). Following these findings, the velocity profiles were averaged over a spanwise length of about 60 m around each concentration profile position.

Furthermore, the $u_*$ values calculated with the method presented in this work are consistent with the $u_*$ values calculated for the entire river section, indicating that the three verticals chosen for the sediment sampling are representative of the mean flow conditions in the cross-section.

We rephrased the section 2.5 in the revised manuscript (p.11, l.18 – 26) to explain why the profiles where averaged over 60 m in the transverse direction.

[Figure]

**Figure RC2_6:** ADCP data analysis at the San Regis station, on the Marañon River (transects measured in April 2012). Bottom screen: bathymetry of the cross-section, with the locus of the sampled concentrations (squares), the vertical sampled (continue lines in black) with the spanwise length of 60 m around each concentration profile position (purple rectangles). Top screen: the shear velocities fitted on each acquisition of a velocity profile (black dots) and their mobile averages over different units of length (color lines).

2) Effect of the velocity profiles averaging on the sediment distributions near river banks.

The cross-sections considered in this study are very large (~from 200 m to 1000 m) and the concentration sampling was performed at three verticals that divided the cross-section according to the river width or the flow rate. Therefore, all the concentration profiles were measured at a distance from the riverbanks greater than 50 m. Then, the velocity profiles averaged over 60 m (i.e. 30 m before and 30 m after the locus of the vertical sampled) were not affected by the riverbanks, same as the concentration profiles (Fig. RC2_6). Furthermore, the flow near the riverbanks can be neglected, given the low velocities and depths in this area.

The section 2.5 has been reworded in the revised manuscript.

*P.12 line 12: The use of the term "random factor" implies some stochasticitiy is incorporated into the model. I'm not sure this is the meaning the authors meant to get across, so i recommend rephrasing to avoid any confusion.*

We agree with the reviewer that the sentence is unclear. We skipped this term in the revised ms, and the section 3.1.2 has been rephrased.

*P.14 line 8: Could the authors provide values as to the "small relvative error" in the text? Is this an RMSE, or other error estimate?*

In the marked manuscript (p.14, l.24 – 26), we have changed "small relative error" for the following sentence: "Overall, the suspension models (Eqs. 7, 9, 11, 12) fit well with the observed profiles (Fig. 6): for 92% of the profiles fitted, the coefficient of correlation (r) were superior to 0.9 and 100% of the r were superior to 0.7".

*P.14 line 17: How likely is it that the velocity at the river bed is zero? Is there completley static flow at the river bed, and if your model predicts this, is it a "reasonable approximation"? Surely there must be flow at the river bed?*

Yes, all the rivers studied here experience a movable bed phenomenon during the high and nominal flows, with a velocity of few cm/s. By "the velocity drops to zero in this zone", we mean: "the velocity decreases rapidly with depth in this zone".

The sentence has been rephrased in the ms (p.15, l.3).

*P.15 line 5: it would be good to have a discussion about the applicability of this model to other systems. The authors say the model is sensitive to Rouse numbers (especially if large). Are there systems where this may be the case and therefore the model is not applicable? Or is this model only really useful on the Amazon Basin?*

It is an excellent proposal. The paragraph p.15, l.23 to p.16, l.2 has been rephrased.
The model can be applied in other large rivers (tropical or not), where the classical theory of suspension applies. The model is especially sensitive to the Rouse number when the index concentration is sampled at the water surface: for large Rouse numbers ($P > 0.3 - 0.4$), the sampling has to be performed in-depth (Fig. 7a), but the model still apply and is accurate. For rivers with Rouse number greater than 0.6, the weight of the velocity distribution in the model

can no longer be neglected as it was in this work (see the assumption, section 2.3.1 and Fig. RC2_3).

Furthermore, the higher the Rouse number, the more the concentration measurement is difficult to perform. Then, the accuracy of the model depends also on the concentration measurement procedure chosen and related uncertainties. These uncertainties depends on the point-sampling integration-time which must be long enough to be representative (Gitto et al., 2017), on the volume of water collected, and on the sampling position(s) defined in the cross-section.

**Figures**

*Figure 1: Could the authors add error bars to show the range of standard of concentrations measured?*

Here the figure with error bars which have been added to the ms.

[Figure]

**New figure 1:** Observed ratios $\alpha = \langle C \rangle / C(h)$ of total mean concentration to the total surface concentration, stacked by river basin, with trend lines. For the Amazonas River basin at TAM, the REG and REQ trend lines were reported. Dashed lines: first bisector.

*Figure 3: I have trouble differentiating the red and blue scale bar here. I would recommend changing to something less of an issue or those of us who are colour blind. I also note that on p12 line 8, figure 3 is described as showing the "sand mass fraction...on the ratio alpha" whereas figure 3 depicts concentration rather than ratio alpha.*

According to referee #1 and #2 reports, Fig.3 is quite difficult to read. We propose the following figure instead, where we can see the increase of $\alpha = \langle C \rangle / C(h)$ with the sand mass fraction $X_s$. The concentration uncertainty and the shear stress variability during the hydrological cycle explain the dispersion observed in this figure.

The following new figure 3 has been added to the ms:

[revised manuscript text omitted]

---

## Author Comment (AC5) · 8 Apr 2019

**Response to review of Kathryn Clark**

William Santini, Benoît Camenen, Jérôme Le Coz, Philippe Vauchel and Jean-Michel Martinez

15/03/2019

We thank Kathryn Clark for her thorough review of our manuscript. We attached a marked version of the manuscript, highlighting all modifications in cyan.

Below we list the referee comments in italic font and our responses in regular font.

Sincerely,

W. Santini et al.

**General Comments**

*I really enjoyed reading this paper and learning about the intricacies of proper River SS concentration sampling in large Amazonian rivers. It is evident that the authors have utilised a vast amount of empirical data coupled with a very comprehensive theoretical overview and application. It is evident that a lot of time and effort went into this paper and it was a pleasure to be able to review it.*

*I imagine this type of paper is aimed at specific scientists who know a lot about the topic already. You might be able to reach a wider audience if you remind readers what the terms mean in the sentence, so they don't have to keep flipping to the terms section. For example, page 11 line 7: possibly repeating what ADCP stands for. Another place a term reminder would help could be in the captions on figures.*

*For example, in the caption for figure 11 it might be good to remind readers what h stands for. Also, there are a lot of stand alone sentences rather than paragraphs throughout the manuscript. Possibly some formatting thought-out could be useful.*

*The tables and figures were very well done.*

Thanks you for this very positive appraisal of our manuscript and the suggestions to reach a wider audience.

1) The corresponding sentences have been rephrased (please report to the marked version of the ms).
   - **P.11, l.16:** "The velocity transects measured with an Acoustic Doppler Current Profiler were used to estimate the shear velocities $u_*$"
   - **P12, l.4 − 12:** This paragraph was reformuled according to the referee #1 comments. These modifications are highlighted in yellow in the ms.
   - **P.12, l.24:** "the index concentrations $C_\chi$ sampled in the upper layers of the flow"
   - **P.12, l.7:** "The measured particle size distributions (PSD)"
   - **P.13, l.16:** "Concerning the whole dataset of fine sediments mean diameters $d_f$"
   - **P.13, l.19:** "The increased sand diameters $d_s$"

- **P.13, l.28 – 29:** "The dataset confirmed that the Zagustin model causes the Rouse number $(P'_\phi)$ to be slightly smaller than that calculated with the Rousean model: $P'_\phi \approx 0.93\, P_\phi$."
- **P.13, l.31 to p. 14, l.1:** "there is a dynamic equilibrium between the settling velocity $w_\phi(d_\phi)$ and the shear velocity $u_*$"
- **P.14, l.7:** "single diffusivity ratio $\beta_\phi = w_\phi / P_\phi \kappa u_*$"
- **P.15, l.1:** the mean concentration $\langle C_\phi \rangle$

2) Figure caption modifications:
- **Caption for figure 2:** "Observed ratios $\alpha = \langle C \rangle / C(h)$ of total mean concentration to the total surface concentration"
- **Caption for figure 3:** "Observed ratios $\alpha = \langle C \rangle / C(h)$ of total mean concentration to the total surface concentration"
- **Caption for figure 5:** "Dimensionless sediment diffusivity coefficient derived from the measured concentration profiles."
- **Caption for figure 8:** "(a) Ratio of sediment to eddy diffusivity $\beta_\phi$ as a function of the ratio $w_\phi / u_*$, with points shaded according to the water level $h$. (b) Idem, after correction of the ratio $(w_\phi / u_*)$. Circles and squares are the mean values of $\beta_s$ and $\beta_f$ calculated per site, respectively."
- **Caption for figure 11:** "(a) Relative error of the predicted $\boldsymbol{\alpha_\phi}$ according to the relative height $\boldsymbol{z_\chi / h}$ of the index sampling, for various Rouse numbers $\boldsymbol{P_\phi}$. (b) Relative error of the concentration sampled, inferred from the Zagustin model and assuming $\frac{\Delta C_\phi}{C_\phi} (\boldsymbol{z_\chi} = \boldsymbol{0.5\, h}) = \pm 10\%$. (c) Relative error on $\langle \boldsymbol{C_\phi} \rangle$ as a function of $\boldsymbol{z_\chi / h}$."

3) The following stand-alone sentences have been modified (please report to the marked version of the ms).

We have suppressed a maximum of stand-alone sentence and tried to keep the text clear.
* * *
*During my PhD I collected water samples at the side of the river in the Madre de Dios river, near the Los Amigos tributary. My team collected a water sample every three hours for two weeks, thus it wasn't possible to go to the middle of the river. How useful are these samples for representing the SS concentration of the river?*

I also collected water samples for sediment concentration determination in the Madre de Dios River in 2013 and 2014, with ADCP data and water levels. I have worked with several local institutions, companies and associations in Puerto Maldonado and I built an hydrological

database with the sparse information's collected in these institutions, satellite data (altimetry), and the sediment monitoring we mounted. Maybe it could be interesting to cross our respective data in the future, to improve the knowledge of the hydro-sedimentary fluxes in this region seriously affected by illegal gold mining.

If your water samples were collected near the riverbank, where the velocity and the water depth are small (and hence where the shear stress is low), the concentration measured are not representative of the cross-section average concentration. However, your index concentrations could be representative of the fine sediments average concentration, although the coarser fraction of the silts tends, as the sands, to settle near the riverbanks. In practice, when the operator collects the water sample from the riverbank, some sediment deposits can be accidentally resuspended, which false the concentration index value. Thus, I recommend to be prudent with the dataset collected.

**Specific comments**

**Abstract**

*Line 22: for the $\alpha$ definition the index and average are switched compared to the terms list*

*Line 26: choice of using washload over fine. Maybe include washload in the terms section*

*Line 31: this model can be coupled, suggested change to could likely, since this wasn't demonstrated in the paper.*

Thanks a lot for these text improvements; the following sentences have been rephrased in the abstract:

- **P.1, l.22:** "between index and average concentrations" was changed for "between depth-average and index concentrations"
- **P1, l.26:** "Two particle size groups, fine sediments and sand"
- **P1, l.32:** "this model can be coupled" was changed for "this model could likely be coupled"

**Introduction**

*Page 2*

*In the first paragraph maybe make specific reference to increase erosion and thus variability in river suspended load?*

*Line 1: possibly include: climate variability, specifically extreme rain events.*

*Line 7: At least in rivers there would be references to help make this statement stronger.*

*Page 3*

*Line 22: I found it quite a challenge to keep up with what all the terms mean. I would let readers know that there is a terms list in the supplement.*

- **P.2:** Thanks for these suggestions. We add the following references in the first paragraph: Martinez et al., 2009; Walling and Fang, 2003, and we rephrased the first paragraph (p.2, l.2 – 14).
- **P.3:** Thanks again, it is done p.3, l.26.

**Materials and Methods**

*Page 5*

*Line 30: should the line read: water level vs discharge and concentration index vs mean concentration?*

*Page 9*

*Line 23: is the fine particle range 0.45 rather than 0.4um?*

- **P.5, l.30:** Yes, thanks you. The sentence was rephrased in the ms, p.6, l.3.
- **P.9, l.23:** Yes, it is 0.45 µm because of the filter's porosity. The sentence was rephrased in the ms, p.10, l.4.

**Results**

*Page 11*

*Lines 22, 23: The grammar of this sentence needs work. If needs a then in the sentence.*

*Page 13*

*Line 25-27: Isn't it the reverse, that it's important to accurately calculate the concentration of the sand fraction because it is so variable? If the authors want to keep the sentence about the fines, then maybe say: in order to accurately capture the small variations in $\varepsilon$, accurate sampling is key.*

*Page 14*

*Line 9: Should this be referring to figure 6 rather than 7?*

*Line 18: Check that this is the proper figures reference for this text.*

*Line 25: suggested grammatical change: very small rather than very low.*

*Line 25/26: suggested grammatical change: Then considering the poor contribution. I would suggest changing it to read small contribution.*

*Line 26: possibly change from low waters to low runoff.*

- **P.11, l.22 − 23:** The section 3.1.1 of the ms has been rephrased (p.12, l.4 − 22).
- **P.13, l.25 − 27:** Because of the low vertical concentration gradients of the fine sediments, it is harder to capture the small variations with depth in $\varepsilon_f$ than in $\varepsilon_s$ (the concentration gradient for the sand fraction being much more pronounced).

   We rephrased the sentence as suggested by the referee (p.14, l.10 − 13): "The diffusivity profiles $\varepsilon_\phi(z)$ were derived from the measured concentration profiles with the discrete

form of Eq. (4). In order to accurately capture the small variations in $\varepsilon_\phi$, accurate sampling is key: the calculation of $\varepsilon_\phi$ requires accurate concentration and sampling height values, particularly for the fine fraction, which experiences low vertical concentration gradients."

- **P.14, l.9:** Yes it is the figure 6. Thanks you.
- **P.14, l.18:** The others figure references are correct in this paragraph.
- **P.14, l.25 − 26:** "poor" was changed for "small" (p.15, l.13)
- **P.14, l.26:** We changed "low water" for "low flow rates" p.15, l.12 of the revised ms.

**Discussion**

*Page 15*

*Line 6: should authors refer to equation 18 when it's not used in the paper?*

*Where is figure 8b referenced?*

*Page 17*

*Line 5: where is the Rouse number compared to particle size?*

*Where is figure 10 referenced?*

*Page 19*

*Line 2: Suggested removal of repeated "the concentration"*

*Line 2: Should "result" read as "results"*

*Line 9: Should qss just read as qs?*

*Page 20*

*Lines 9-12: If this is the only statement made about the usefulness of remote sensing, I would modify the text in the abstract.*

- **P.15, l.6:** If the index concentration is sampled in the central region of the flow [0.2 h, 0.8 h] and not at the water surface, the Eq. 18 is applicable (Fig. 7a) and simpler than Eq. 17.
- **Figure 8b** was referenced in the text p.17, l.27.
- **P.17, l.5:** We mean here that in the Amazonian context, the Rouse number variability is controlled at first order by the shear velocity: Therefore, the particle size is a second order factor.

  We rephrased the sentence p.18, l.3 − 5: "This result shows that the shear velocity mainly controls the Rouse number variability in a given site (Fig. 9b). The variations in particle size ($d_\phi$ was considered as a constant for all the hydrologic conditions in the current study) are therefore a second order factor."
- **Figure 10** was referenced in the text p.18, l.24 − 25.

- **P.19, l.2:** We rephrased the sentence to avoid the repetition: "For instance, the concentration at $z_{\chi_1} = 0.7\ h$ and at $z_{\chi_2} = 0.3\ h$" (p.20, l.2 − 3).
- **P.19, l.2:** Thanks. It was corrected in the marked ms (p.20, l.3).
- **P.19, l.9:** It is $q_{s\phi}$ as in Eq. 22. Thanks again for this typo. The ms has been modified p.20, l.10.
- **P.20, l.9 − 12:** We refer to the remote sensing technics for sediment concentration monitoring in many sections of the manuscript (abstract, introduction, method and discussion). It is one of the motivations of our study. Indeed, remote sensing technics give only access to a surrogate of the mean concentration and must be coupled with hydraulic modeling or/and correlated with in situ data to be a quantitative measurement and not only a qualitative information.

**Appendices**

*Page 35*

*Line 13: should the fines be from 0.45 to 0.63um?*

- **P.35, l.13:** Yes, it is 0.45 µm because of the filter's porosity. The term list was modified (p.38, l.8 − 9).

**Additional references:**

[revised manuscript text omitted]